# From Co-Infections to Autoimmune Disease via Hyperactivated Innate Immunity: COVID-19 Autoimmune Coagulopathies, Autoimmune Myocarditis and Multisystem Inflammatory Syndrome in Children

**DOI:** 10.3390/ijms24033001

**Published:** 2023-02-03

**Authors:** Robert Root-Bernstein

**Affiliations:** Department of Physiology, Michigan State University, East Lansing, MI 48824, USA; rootbern@msu.edu

**Keywords:** neutrophils, SARS-CoV-2, cytokine storm, bacterial co-infection, LPS, Kawasaki disease, MIS-C, Toll-like receptors, NOD-like receptors, thrombosis

## Abstract

Neutrophilia and the production of neutrophil extracellular traps (NETs) are two of many measures of increased inflammation in severe COVID-19 that also accompany its autoimmune complications, including coagulopathies, myocarditis and multisystem inflammatory syndrome in children (MIS-C). This paper integrates currently disparate measures of innate hyperactivation in severe COVID-19 and its autoimmune complications, and relates these to SARS-CoV-2 activation of innate immunity. Aggregated data include activation of Toll-like receptors (TLRs), nucleotide-binding oligomerization domain (NOD) receptors, NOD leucine-rich repeat and pyrin-domain-containing receptors (NLRPs), retinoic acid-inducible gene I (RIG-I) and melanoma-differentiation-associated gene 5 (MDA-5). SARS-CoV-2 mainly activates the virus-associated innate receptors TLR3, TLR7, TLR8, NLRP3, RIG-1 and MDA-5. Severe COVID-19, however, is characterized by additional activation of TLR1, TLR2, TLR4, TLR5, TLR6, NOD1 and NOD2, which are primarily responsive to bacterial antigens. The innate activation patterns in autoimmune coagulopathies, myocarditis and Kawasaki disease, or MIS-C, mimic those of severe COVID-19 rather than SARS-CoV-2 alone suggesting that autoimmunity follows combined SARS-CoV-2-bacterial infections. Viral and bacterial receptors are known to synergize to produce the increased inflammation required to support autoimmune disease pathology. Additional studies demonstrate that anti-bacterial antibodies are also required to account for known autoantigen targets in COVID-19 autoimmune complications.

## 1. Introduction

### 1.1. Conceptual Framework for This Review

In previous publications, I have reviewed evidence that autoimmune diseases are characterized by increased inflammation due to the synergistic activation of multiple innate immune system receptors [1]. I have also reviewed studies suggesting that cytokine over-production syndromes, such as the so-called “cytokine storms” associated with severe COVID-19, acute lung injury (ALI), acute respiratory distress syndrome (ARDS) and sepsis may be the consequence of similar synergistic activation of innate immune system receptors [2]. In both cases, I argued that the observed sets of activated receptors can only be accounted for by synergies between multiple antigens, most frequently derived from virus–bacteria pairs known to be associated with each disease or syndrome.

This paper explores the ways that COVID-19 cytokine over-production syndrome sets the stage for autoimmune disease sequelae and presents evidence that COVID-19-associated autoimmunity is unlikely to be explained by SARS-CoV-2 infection by itself, but requires concomitant stimulation either by bacterial co-infections or release of self-antigens due to tissue damage caused by the virus. Two types of autoimmune disease associated with COVID-19 are used as case studies: the vascular/myocardial Kawasaki-Disease-like syndrome Multisystem Inflammatory Syndrome in Children (MIS-C), sometimes called Pediatric Inflammatory Multisystem Syndrome Temporally Associated with SARS-CoV-2; and COVID-19 autoimmune coagulopathies resembling thrombotic thrombocytopenia and antiphospholipid syndrome. A related issue that will be addressed is the mechanisms that may underlie the neutrophilia and production of neutrophil extracellular traps (NETs) that characterize both severe COVID-19 and its autoimmune complications.

This review is structured as follows. This first section will provide a framework for aggregating, comparing and analyzing the results of studies that characterize innate receptor activation patterns. These patterns will be used as analytical tools in later sections. An overview of innate immune system receptor responses to viral and bacterial antigens is followed by a summary of known innate receptor synergisms and antagonisms, and their relationship to hyperinflammatory conditions, such as those found in severe COVID-19. Evidence of hyperinflammation in severe COVID-19 and its various autoimmune complications will then be reviewed to set the context for examining the role of innate receptor activation in these syndromes.

The second section will review what is known about innate receptor activation resulting from exposure to SARS-CoV-2, its protein antigens and the adenovirus vectors used to deliver some COVID-19 vaccines. These activation patterns will be compared with those found in severe COVID-19 patients and patients suffering from COVID-19-related autoimmune complications. The roles of neutrophilia and NETs in these diseases will then be reviewed. Since neutrophilia and NET production are highly associated with the presence of bacteria and/or specific damage-associated molecular pattern (DAMP) activators, the innate receptor activation patterns of COVID-19-associated bacterial infections and DAMPs will also be reviewed. This reviewed literature is then summarized to address the question of whether SARS-CoV-2 is sufficient to explain the innate activation patterns found in severe COVID-19 and its autoimmune complications. The extent to which the autoimmune activation patterns in COVID-19 and its complications differ from each other is also examined. This review demonstrates clearly that SARS-CoV-2 is not sufficient to induce the innate receptor activation patterns characterizing severe COVID-19 or its autoimmune complications.

The third section of the review integrates the information reviewed in the second section within the analytical framework of receptor synergisms developed in the first section. Receptor synergism patterns will be used to explore whether SARS-CoV-2 antigens combined with various bacteria and/or host antigens (DAMPs) are capable of explaining the innate receptor activation patterns characterizing severe COVID-19 and its autoimmune complications. This section also directly addresses possible mechanisms to explain why severe COVID-19 and its autoimmune complications are characterized by hyperinflammatory syndromes, and also the possible origins of very rare autoimmune complications following COVID-19 vaccinations.

The concluding section of this review will address the novel diagnostic, therapeutic and experimental implications of the preceding synthetic analysis. It will emphasize the need to vaccinate against, test for and treat possible co- or super-infections of SARS-CoV-2. Additionally, it will outline possible animal models that might be set up to explore how SARS-CoV-2 (and adenovirus vectors) may interact with such co- or super-infections, or with DAMPs released during severe SARS-CoV-2 infections, to produce the hyperinflammatory conditions underlying severe COVID-19 and its complications.

### 1.2. Overview of Innate Immune System Receptors

Within the innate immune system, macrophages, neutrophils and dendritic cells identify and eliminate pathogens that might cause infection. These cells are activated by interaction between antigens and a variety of pattern-recognition receptors, generally classified into Toll-like receptors (TLRs), nucleotide-binding oligomerization domain- (NOD)-like receptors (NLRs), C-type lectin receptors (CLRs) and retinoic acid-inducible gene I (RIG-1)-like receptors (RLRs). These receptors recognize conserved molecular structures of pathogens called pathogen- (or microbe-) associated molecular patterns (PAMPs or MAMPs) and can also respond to host-released DAMP antigens. Figure 1 summarizes some of the better-known PAMPs associated with the TLRs, NLRs and RLRs that are the focus of the present study. These innate receptors were selected for analysis because of the abundance of available research on the activation of these receptors in COVID-19 and its associated autoimmune diseases. For simplicity, CLR activation has been omitted because it involves a large number of receptor types that can be activated by glycans on both viruses and bacteria, creating a very complex activation picture that will need to be addressed separately; moreover, there is substantially less information available for CLRs with regard to COVID-19 and its autoimmune complications than for TLRs, NLRs and RLRs (reviewed in [1,2]).

Several general principles can be drawn from Figure 1. One is that viruses generally activate TLR3, TLR7, TLR8, TLR9, RIG-I and/or MDA5, all of which are intracellular receptors present either in the cytosol or within endosomes. Bacterial antigens, in contrast, are generally recognized by TLR1, TLR2, TLR4, TLR5 and TLR6—all of which are located in the cell membrane—and NOD1 or NOD2, which are cytoplasmic [3]. Some of these receptors normally dimerize so that TLR2 is often found as a complex with TLR1 or TLR6, activating both sets of receptors simultaneously. In addition, the inflammasome NLRP3 (which is not shown in Figure 1) consists of a multi-protein complex involving multiple receptor types that can be stimulated by adjuvants such as alum, fungal antigens such as zymosan, bacterial toxins and mitochondrial antigens [4]. DAMP antigens can also activate these innate receptors, often as a result of molecular mimicry of corresponding PAMPs [1,4].

The types of cytokines released as a result of TLR, NLR and RIG1 activation depend on the sets of receptors activated by PAMPs and DAMPs. For example, TLRs 3 and 6 can activate the Toll/IL-1-receptor-domain-containing adaptor-protein-inducing interferon β activators (TRIF) pathway, resulting in interleukin production and a Th1 (or cellular) immune response. The other TLRs activate the myeloid differentiation primary response protein 88 (MyD88) pathway, which results in the production of proinflammatory cytokines, such as TNFα and IL6, and the production of a Th2 (antibody) response. Notably, TLR4 can participate in the activation of both the TRIF and MyD88 pathways, making TLR4 a critically important player (and one that will repeatedly reappear below) in any syndrome in which multiple types of cytokines are induced. NLRPs (including NLRP3) mediate the assembly of inflammasome complexes, leading to the activation of procaspase-1 and the release of ILβ and IL18. RIG1, NOD1 and NOD2 mediate the assembly of complexes that mediate MyD88 activation via the mitogen-activated protein kinase (MAPK) and nuclear factor kappa-light-chain-enhancer of activated B cells (NF-κB) signaling pathways. NLR and RLR activation results in the release of type 1 interferon, as well as the process of cellular autophagy (reviewed in [1,2]). The key point here is that in order to activate both the MyD88 and TRIF pathways, and thus to induce both interferons and interleukins, a complex interplay of multiple innate receptors associated with both viral and bacterial antigens, or self-antigens that mimic both, is necessary. TLR4 (which is activated by bacterial lipopolysaccharides, such as LPS) is a key receptor in this interplay. As will be demonstrated below, cytokine activation patterns in severe COVID-19 and its autoimmune complications invariably involve both the MyD88 and TRIF pathways simultaneously, suggesting roles for self- or bacterial antigens in addition to those presented by the SARS-CoV-2 virus.

### 1.3. TLR and NLR Synergisms and Antagonisms

A key principle that has underpinned my previous explorations of innate immune system hyperactivation is that most TLRs, NLRs and RLRs are known to synergize with, or antagonize, some subset of the others. Since I have recently reviewed and integrated hundreds of studies from the relevant literature [1,2], I will not do so again here, but simply summarize the key interactions in Figure 2, which is supplemented by the original references appended to the end of the References section. In general, when antigens are presented concurrently to innate receptors that can synergize, they up-regulate both pairs of receptors in a more-than-additive manner; however, synergistic pairs of receptors may down-regulate each other if one antigen is presented significantly prior to (usually several days or more) the other. Additionally, some receptor pairs appear to interact only antagonistically, so that activation of one down-regulates the other(s). As Figure 2 illustrates, a significant number of innate receptor synergisms and antagonisms have been documented, creating a very complex regulatory system. Notably, TLRs generally associated with viral antigens mainly synergize with TLRs associated with bacterial antigen activation. Virus–virus synergies and bacteria–bacteria synergies are much less likely.

The layout of Figure 2 has been designed to show all the TLR–NLR–RLR interactions known at this time, and will be used as a template for analyzing what synergies and antagonisms can be predicted from the known activation patterns of SARS-CoV-2, its bacterial co- or super-infections, and the DAMPs released during COVID-19 tissue damage. The boxes with plus or minus signs in Figure 2 were generated by studies that utilized well-characterized TLR, NLR and/or RLR agonists. As can be seen in the Figure, not every TLR, NLR or RLR interacts with others, and the absence of an interaction is designated by blank boxes. Different sets of TLR, NLR and/or RLR agonists create different patterns of interactions. For example, TLR4 agonists, such as bacterial lipopolysaccharides (LPS), synergize with almost all virus-antigen stimulated TLRs, NLRs and RLRs. TLR6, in contrast, is activated by bacterial lipoproteins but is known to synergize only with TLR agonists, such as bacterial glycoproteins. Such interaction patterns are at the heart of the analysis presented below, and the same format illustrated in Figure 2 will be used to display the data for each disease or syndrome analyzed.

### 1.4. Hyperinflammation, Bacterial Co- and Super-Infections and Autoimmunity in COVID-19

Increased inflammation involving the activation of a wide range of TLRs, NLRs and RLRs characterizes severe cases of COVID-19 and its autoimmune sequelae, and differentiates these from milder cases that do not experience autoimmune complications. COVID-19 is generally described as a disease caused by SARS-CoV-2 and its symptoms are generally mild to moderate in the vast majority of cases. General symptoms often consist of fever, chills, dyspnea, sore throat, myalgia, congestion or rhinitis, dysgeusia and/or anosmia, and sometimes nausea, vomiting or diarrhea (https://www.cdc.gov/coronavirus/2019-ncov/symptoms-testing/symptoms.html, accessed 26 September 2022). Severe COVID-19 differs radically from asymptomatic and mild cases in many ways, including presentation with an atypical cytokine storm characterized by lymphopenia, neutrophilia and an increased neutrophil-to-lymphocyte and neutrophil-to-CD8+ T cell ratio; increases in concentrations of inflammatory cytokines produced from both the TRIF and MyD88 pathways, including IL-6, TNF-α, IL-1β, IL-8, IL-10, and IL-18; and production of neutrophil extracellular traps [13,14,15]. All of these symptoms are typical of bacterial infections or bacterial super-infections of viral pneumonias [16,17]. Such cytokine over-production is consistent with septic shock and acute respiratory disease syndrome (ARDS) being the most common complications observed in severe COVID-19 cases [18].

The role of bacteria in driving the severity of COVID-19 has been well documented, and some of the antigens are known to stimulate increased inflammation. For example, the concentrations of the bacterial antigens (1→3)-β-D-glucan (BG) and lipopolysaccharide (LPS) [19], as well as 16S ribosomal (bacteria-specific) RNA [20] in serum, correlate directly and very significantly with COVID-19 severity. Recent reviews have documented the incidence of SARS-CoV-2 co-infections to be between 42 and 94%, depending on whether the studies examined hospitalized COVID-19 patients or those admitted specifically to intensive care, and whether isolation or PCR was used for diagnostic purposes [21,22,23,24,25,26]. *Streptococcus pneumoniae* and *Staphylococcus aureus* were the most common bacteria identified in most studies, followed by *Klebsiella pneumoniae*, *Haemophilus influenzae*, *Mycoplasma pneumoniae*, *Acinetobacter baumannii*, *Legionella pneumophila* and *Chlamydia pneumoniae* [21,22,23,24,25,26]. Co-infections with SARS-CoV-2 and *Pneumocystis jirovecii* or *Legionella pneumophila* have also been reported [26], and co-infections with adenoviruses or human rhinoviruses were documented in 5 -7.3% of severe COVID-19 cases [24,25]. Notably, the atypical cytokine storm associated with severe COVID-19 can be moderated by treatment with antibiotics prior to intensive care unit admission, or exacerbated if treatment for bacterial co-infection is delayed to the mid-to-late phase of the disease [27]. Thus, a role for bacterial co- or super-infections in severe COVID-19 is very likely in most cases.

Severe COVID-19 is also distinguished from mild and asymptomatic cases by often being complicated by a range of autoimmune diseases [28,29,30,31,32,33,34,35,36,37]. These autoimmune complications include, but may not be limited to, Grave’s disease (anti-thyroid); type 1 diabetes mellitus (anti-pancreatic beta cells); neurological syndromes mimicking Guillain–Barré syndrome, as well as possibly Parkinson’s disease, multiple sclerosis and narcolepsy; autoimmune hepatitis; IgA nephropathy; rheumatoid arthritis; systemic lupus erythematosus; a range of autoimmune coagulopathies, including immune thrombotic thrombocytopenia and an anti-phospholipid-like syndrome; autoimmune myocarditis and endocarditis; postural orthostatic tachycardia syndrome; and the Kawasaki-disease-like vasculitis syndrome, MIS-C. These autoimmune diseases have also been associated—though at much lower rates of incidence than in severe COVID-19 cases—with SARS-CoV-2 vaccination [36,37]. Among the mysteries that attend our ignorance concerning the initiators of these COVID-19-associated autoimmune diseases is how a single virus, SARS-CoV-2, could be capable of initiating such a wide range of autoimmune symptoms [31]. One of the goals of this paper is to provide a possible explanation for this mystery.

As noted above, this paper will focus primarily on SARS-CoV-2-related autoimmune coagulopathies and vascular/myocardial autoimmunity as a way to unravel some of the mysteries about the relationship between the hyperinflammatory state associated with severe COVID-19 and its associated autoimmune diseases. Both Kawasaki disease (KD) and its COVID-19 variant, MIS-C, are post-infectious, immunologically mediated forms of vasculitis characterized by cardiac involvement and a wide range of symptoms, including fever, rash, cervical lymphadenopathy, neurological symptoms, thrombocytopenia and septic shock, sometimes accompanied by conjunctival and gastrointestinal inflammation [38,39,40,41]. However, KD and MIS-C can be differentiated in a number of key ways: the infectious trigger(s) of KD are unknown, with various viral (most statistically significantly coronaviruses, adenoviruses and respiratory syncytial virus [42,43,44]) and bacterial (including Staphylococci and Streptococci [45,46,47]) infections being associated temporally with its onset, whereas MIS-C clearly follows 3 to 6 weeks after SARS-CoV-2 infection; KD is more common among children of Asian descent, while MIS-C is most common in children of African or Hispanic heritage; coagulopathies are common in MIS-C and less common in KD, while conjunctival and mucus membrane involvement is more common in KD than in MIS-C. Immunologically, neutrophilia, lymphopenia, an increased neutrophil-to-lymphocyte ratio and cytokine over-production are common in MIS-C, and KD shares neutrophilia as a common problem, but lymphopenia and cytokine storms are more rare in KD [38,39,40,41,48]. However, MIS-C does share many clinical commonalities with severe COVID-19, including the presence of co-infections with Streptococcal and Staphylococcal bacteria [39,40,41,48], presence of coagulopathies, increases in C-reactive protein (CRP), ferritin, procalcitonin, D-dimer and fibrinogen, an increased prothrombin time and over-production of IL-2, IL-6, IL-8 and TNFalpha [38,39,40,41,48,49,50]. Significantly, shock associated with bacterial sepsis, sometimes mimicking toxic shock syndrome, is also common in MIS-C, another clinical symptom shared with severe COVID-19 [49, 51] (Table 1). KD shares many of these clinical marker changes as well, including a direct correlation between inflammatory markers such as neutrophilia, anti-cardiolipin antibodies, increased D-dimer, CRP and ferritin, and the severity of coronary artery lesions and resistance of the disease to therapy [52,53,54]. As in MIS-C and severe COVID-19, the vast majority of KD patients (77% in one study [41]) exhibited evidence of unusually high levels of microbe-associated molecular pattern antigens (MAMPs) characteristic of *Streptococci* [39], *Bacillus cereus, Yersinia pseudotuberculosis* and *Staphylococcus aureus* [55] in their blood serum. Thus, some investigators have argued that KD is a reasonable model for helping to understand the etiology and pathogenesis of MIS-C [51,56,57]. Also noteworthy is the observation that KD and MIS-C often share myocardial complications, including autoimmune myocarditis (AM), which is predicted by the high C-reactive protein (CRP) and troponin found in the blood samples of many patients [58]. Prior to the COVID-19 pandemic, AM was associated with combined infections involving, most frequently, coxsackie- or other enteroviruses, in combination with bacterial co-infections, most commonly group A streptococci [59,60]. As in KD and MIS-C, AM is characterized by the presence of circulating immune complexes and cytokine over-production [1,58,61,62].

Table 1, and in particular the data concerning rates of ischemic stroke, fibrinogen and D-dimer up-regulation, also reveal that autoimmune coagulopathies are common among severe COVID-19 patients [63,64,65,66,67]. The clinical symptoms of these COVID-19 complications bear many similarities to pre-COVID-19 coagulopathies, as well as some differences. Approximately two-thirds [68] of patients with severe or critical COVID-19 develop arterial and venous thromboses that increase their probability of death, as compared with mild cases. For comparison, the prevalence of such complications among the general population is about 40 to 50 per 100,000 [69]. Coagulation abnormalities include increased fibrin formation and D-dimer, prolonged activated partial-thromboplastin time (aPTT), thrombocytopenia, microclotting, deep-vein thrombosis (DVT) and/or arterial thrombosis reminiscent of antiphospholipid syndrome (APS) [68]. Autoantibodies against phospholipids (PL), such as cardiolipin (CL), and various phospholipid-binding proteins (aPL), such as lupus anticoagulant (La), β2 glycoprotein I (β2GPI) and phosphatidylserine/prothrombin (aPS/PT), promote thrombosis, but, unlike APS, are not diagnostic for it. Antibodies against at least one of these antigens were found in 52% of hospitalized COVID-19 patients [70,71] and about 75% of severe and critical patients [68]. The presence of individual lupus anticoagulant (La) antibodies correlated poorly or not at all with thrombotic risk [70], the development of APS in COVID-19 patients being correlated with very high levels of such autoantibodies [70] (which is correlated in turn with unusually high SARS-CoV-2 antibody titers [70]), cytokine over-production [68], the presence of two or more autoantibodies [72,73,74] and formation of circulating immune complexes (CIC) [75,76]. Anti-platelet factor 4 (PF4) antibodies have also been found to correlate highly with a syndrome mimicking anti-heparin-PF4-induced thrombocytopenia in COVID-19 [77,78], which has also occurred rarely following SARS-CoV-2 vaccination [74,79,80]. Since severe COVID-19 is highly associated with bacterial co-infection, it is notable that prior to COVID-19, autoimmune coagulopathies were correlated with infections involving *Streptococci*, *Staphylococci*, *E. coli*, *Clostridia* and a number of other bacteria exhibiting antigens that mimic blood coagulation proteins (reviewed in [17,74,80,81,82,83,84,85]). A number of viral infections, including human immunodeficiency virus, hepatitis B virus and hepatitis C virus, have also been highly associated with development of anti-CL antibodies in 20 to 50% of infected patients [83], but as in severe COVID-19, only a few of these patients developed APS. Those developing were characterized by the presence of both anti-CL and anti-β2GPI antibodies rather than either antibody alone [84,85], a point of significance with regard to possible etiologies of the syndrome, and one to which this review will return below.

In sum, the evidence suggests that autoimmune complications associated with SARS-CoV-2 have appeared mainly among those with the most severe COVID-19 symptoms; that severe COVID-19 cases differed from mild and moderate cases in having a much higher incidence of bacterial co-infections and the presence of bacterial antigens in blood samples; that such co-infections were, in turn, associated with increased inflammation characterized by neutrophilia, production of neutrophil extracellular traps (NETs), lymphopenia and cytokine over-production; and that all of these features were also characteristic of the autoimmune diseases such as KD, APS and autoimmune myocarditis that emerged in new forms with the COVID-19 pandemic (Table 2). The purpose of this research is to use two sets of autoimmune disease complications associated with severe COVID-19—vascular/myocardial autoimmunity and autoimmune coagulopathies—as case studies to test the hypothesis that the increased inflammation observed in COVID-19 autoimmune diseases is a result of activation of synergistic sets of innate immune system receptors by SARS-CoV-2 and bacterial antigens. It is proposed that such microbial synergisms are particularly necessary to explain the neutrophilia and NET production, which are classically associated with bacterial or combined infections, rather than uncomplicated viral ones.

The hypothesis that autoimmune complications in COVID-19 follow synergistic activation of TLRs, NLRs and RLRs necessitates analyzing which of these receptors are activated by SARS-CoV-2, and whether this activation is sufficient to explain the range of innate receptors activated during severe COVID-19 or its autoimmune complications. To this end, the next section reviews current studies of innate receptor activation in human beings exposed to SARS-CoV-2 antigens and adenoviruses, such as those used as vectors for some COVID-19 vaccines. The innate receptor activation patterns of these viruses are then compared with those observed in severe COVID-19, MIS-C, Kawasaki disease, autoimmune myocarditis and autoimmune coagulopathies associated with COVID-19.

## 2. TLR and NLR Activation in SARS-CoV-2, Severe COVID-19 and Its Autoimmune Complications

### 2.1. Overview

Innate receptor activation patterns for SARS-CoV-2 and its spike protein, adenovirus and the various autoimmune diseases associated with severe COVID-19 are reviewed in this section. The results strongly suggest that autoimmune complications of COVID-19 are very likely due to synergisms between viral and bacterial (or possibly fungal) antigens. SARS-CoV-2, its spike protein and adenovirus vectors for SARS-CoV-2 vaccines only activate TLR3, TLR7, TLR8, RIG-1, MDA-5 and NLRP3—the receptors typically stimulated by viral antigens—whereas each of the autoimmune diseases associated with COVID-19 are characterized by activation of not only these virus-activated receptors but also TLR2, TLR4, TLR5 or TLR6 and NOD1 or NOD2. The latter innate receptors are usually activated by bacterial or fungal antigens or host antigens (DAMPs) that mimic them.

### 2.2. Innate Receptor Activation by SARS-CoV-2 and Its Vaccines

First, SARS-CoV-2 activation of innate receptors will be considered. Some investigators have reported that SARS-CoV-2 proteins activate TLR2 [86,87] and, in particular, that the SARS-CoV-2 spike protein, which is the basis for most SARS-CoV-2 vaccines, is the activating protein [88,89]. Zheng [86], however, found that the spike protein did not bind to TLR2, but that TLR2 activation was due to the envelope protein. Not only are these results inconsistent, but other studies found no evidence of TLR2 activation by any SARS-CoV-2 protein or the whole virus (reviewed in [90]). The contradictory results have been clarified by Cinquegrani et al. [91] and Ouyang et al. [92], who demonstrated that reports of TLR2 activation were due to contamination with bacterial or yeast antigens present as a result of using recombinant genetic engineering techniques to produce the proteins in *E. coli* or *Saccharomyces cerevisiae*. TLR2 activation by SARS-CoV-2 or its spike protein is not, therefore, included in the analysis that follows.

SARS-CoV-2 activation of TLR4 is also questionable. Modeling and binding studies have suggested that some SARS-CoV-2 proteins, particularly the spike protein, may bind to TLR4 [93,94], but ex vivo experiments by Bartolotti et al. [95] failed to find any activation of TLR4 by SARS-CoV-2. Once again, the studies of bacterial or yeast antigen contamination of SARS-CoV-2 proteins produced by recombinant means are relevant [90,91], and, once again, in light of contradictory evidence and evidence of contaminants, TLR4 activation by SARS-CoV-2 is not included in the analysis that follows.

Consistent evidence exists for SARS-CoV-2 activation of TLR3 [95,96,97], TLR7 [87,93,95,98,99], NLRP3 [100,101] and MDA-5 [102,103,104,105]. Several investigators report additional activation of RIG-1 [97,103,104] and TLR8 [98,99,106], but in both cases, Bortolotti et al. [95] reported finding no activation. Yin et al. [102] have also reported NOD1 activation by SARS-CoV-2, but this may once again be due to bacterial contamination [91,92] and needs further investigation. In sum, other than the questionable data concerning TLR2 and TLR4 activation, the SARS-CoV-2 spike protein has only been reported to activate TLR7 [89] and NLRP3 [107,108].

Adenoviruses, which are used as vectors for the SARS-CoV-2 spike protein in some vaccines, have only been reported to activate TLR9 [109,110,111,112], with experiments consistently showing no activation of TLR2, TLR4, RIG-1, MDA-5 or NLRP3 [111,112,113,114]. Oddly, there seems to be no data on possible TLR3, TLR7 or TLR8 activation, so their participation in innate activation cannot be excluded.

In sum, the best available data suggest that SARS-CoV-2 activates a majority of the virus-associated innate receptors, including TLR3, TLR7, MDA-5 and NLRP3, as well as perhaps TLR8 and RIG-1, while the major components of most SARS-CoV-2 vaccines—the SARS-CoV-2 spike protein and adenoviruses—activate TLR7 and TLR9, respectively, and probably NLRP3 as well. There is no reliable evidence that innate receptors that mainly respond to bacterial antigens, such as TLR2, TLR4, TLR5, TLR6, NOD1 or NOD2, are activated by adenoviruses, SARS-CoV-2 or its spike protein.

### 2.3. Innate Receptor Activation in Severe COVID-19

Innate receptor activation profiles associated with severe COVID-19 are strikingly different than those characterizing SARS-CoV-2 infections or SARS-CoV-2 vaccination. As would be expected from the SARS-CoV-2 viral activation pattern just described, TLR3 [20,115], TLR7 [20,115,116,117,118], TLR8 [20,56,86,115,117,118,119], TLR9 [86,115], NLRP3 [101,118,120,121], RIG-1 [122] and MDA-5 [122] are activated in patients with severe COVID-19. The activation of TLR9 is notable for the fact that SARS-CoV-2 itself has not been shown to activate TLR9 but adenoviruses do, suggesting that some severe COVID-19 patients may be co-infected with adeno- or other viruses in addition to SARS-CoV-2. More importantly, additional innate activation usually associated with bacterial antigens has also been documented repeatedly in severe COVID-19, which cannot be accounted for by SARS-CoV-2 or other viruses. These additional activations involve TLR1 [86,117], TLR2 [86,118,123,124], TLR4 [20,86,117,123,124], TLR5 [20,86], TLR6 [125], NOD1 [126] and NOD2 [127]. These studies have also demonstrated that these receptors were not activated in mild cases of disease, suggesting that bacterial (or fungal) co- or super-infections play important roles in stimulating cytokine over-production in severe COVID-19 cases.

### 2.4. Innate Receptor Activation in MIS-C and KD

The pattern of innate receptor activation in MIS-C is much more similar to that of severe COVID-19 than to uncomplicated SARS-CoV-2 by itself. However, there are significantly less data available for MIS-C than for COVID-19 more generally, making this conclusion more tentative than could be desired. MIS-C patients have been reported to have up-regulated TLR2 [56,128], TLR3 [129,130,131], TLR4 [130], TLR6 [131], TLR7 [56,129,132], TLR8 [56], NLRP3 [57], RIG-1 [128] and MDA-5 [128]. There appear to be no studies of TLR1, NOD1 or NOD2 activation in MIS-C thus far. This activation pattern once again suggests stimulation by both viral and bacterial antigens.

Since KD bears many similarities to MIS-C and substantially more research has been carried out into innate activation in KD, the innate activation pattern for KD may also shed light on the etiology and pathogenesis of both diseases. Once again, the KD pattern is indicative of activation by a combination of both viral and bacterial antigens. Up-regulation has been observed for TLR1 [133,134], TLR2 [56,133,134,135,136,137,138], TLR4 [56,133,134,139], TLR5 [133,134,140], TLR6 [141], TLR8 [133,134,142] (but no change reported in [137]), TLR9 [143], NOD1 [144], and NLRP3 [57,145,146,147]. TLR3 [134,137] and TLR7 [56,142] are down-regulated and NOD2, RIG-1 and MDA-5 do not appear to be stimulated in chronic KD, though they may be transiently activated during the precipitating infection(s). Though both KD and MIS-C share much the same pattern of innate receptor up-regulation of both virus and bacteria-activated receptors, lack of stimulation of TLR3, TLR7, RIG-1 and MDA-5 activation in KD may differentiate the two diseases and suggest that KD may be triggered by a virus with a much different innate immune system activation pattern than SARS-CoV-2.

### 2.5. Innate Receptor Activation in Autoimmune Myocarditis

Severe COVID-19, KD and MIS-C are often complicated by autoimmune myocarditis (AM) so that a comparison between the innate activation patterns of AM in relation to these syndromes is warranted. Unfortunately, very little research has been carried out regarding human AM (as opposed to animal models, which are well characterized [1]). Only one, very limited, study of AM in severe COVID-19 appears to have been carried out so far [147], which found activation of TLR4 and TLR9, suggesting both bacterial and viral triggers. Studies of innate activation unrelated to COVID-19 are also very limited [148,149,150,151,152], revealing TLR1, TLR2, TLR3, TLR4, TLR7 and TLR8 activation. No data could be found concerning NOD1, NOD2, NLRP3, RIG-I or MDA-5 in human patients. Two damage-associated molecular pattern (DAMP) activators have also been identified for AM, which are myosin and cardiolipin (CL), the former activating TLR2 and TLR8 [153]; the latter, TLR2, TLR4 and NLRP3 [154,155,156,157,158,159,160]. Although these data are limited, in aggregate they suggest that the various forms of AM all share the characteristic of activating both virus-associated (TLR3, TLR7, TLR8 or TLR9) and bacteria-associated (TLR2 and TLR4) innate receptors with severe COVID-19, KD and MIS-C. However, further research is clearly called for to clarify the roles of innate receptor activation in both non-COVID-19 AM and COVID-19 AM.

### 2.6. Innate Receptor Activation in Autoimmune Coagulopathies

The presence of CL as a DAMP [161] in cardiac autoimmunity is interesting in a different COVID-19 context, as CL is also a common cell-membrane component of pathogenic bacteria [162,163,164], so it can also act as a PAMP activator, inducing autoantibodies, by mimicking host CL. Such CL autoantibodies have been found as one aspect of anti-phospholipid syndrome (APS), which sometimes accompanies COVID-19 MIS-C and AM, and sometimes appears separately from these other autoimmune complications [165,166,167,168,169]. Anti-CL antibodies (aCL), complexed with CL, are themselves stimulators of innate immunity, activating TLR4 but not TLR2 [170].

Innate activation in APS is much broader than can be accounted for from CL. Anti-CL antibodies (aCL) are associated with lupus anticoagulant (La), which targets phospholipids. Both often co-occur with anti-β2GPI (which is a phospholipid and CL-binding protein) and anti-phosphatidyl serine/prothrombin [aPS/PT] antibodies [171,172,173,174]. While individual antibodies against these human proteins have been documented frequently among COVID-19 patients, individually they have not been found to correlate well with either severity or risk of thrombosis [171,172,173,174]. However, combinations of these antibodies predict COVID-19 severity and increase thrombosis risk [171,172,173,174]. No information was found directly studying La interactions with innate receptors, but many studies have characterized overall receptor activation in APS and more specifically for β2GPI, thrombin and their respective antibodies.

APS is characterized by the activation of TLR1 [175], TLR2 [175,176,177], TLR4 [170,176,178,179], TLR6 [175], TLR7 [180,181,182], TLR8 [181,182,183], NOD2 [184] and NLRP3 [185]. β2GPI and its antibodies can account for some of this activation. β2GPI binds to and activates TLR1 [186], TLR2 [186,187,188], TLR4 [186,188,189,190] and TLR6 [186], while antibodies against β2GPI (usually in complex with β2GPI) activate TLR4 [178,191,192,193,194] and NLRP3 [185]. Notably, aPS/PT are not known to activate innate receptors, but thrombin blocks agonist binding to TLR2 and TLR4 [195], suggesting that aPS/PT may bind to TLR2 and TLR4 as well but act as an antagonist. In short, APS is, like MIS-C, KD and AM, characterized by activation of both virus-associated and bacteria-associated receptors.

### 2.7. Comparison of Innate Receptor Activation Patterns in COVID-19-Related Diseases

The previous results are summarized in Figure 3, which clearly shows that SARS-CoV-2 activates mainly virus-associated innate receptors TLR3, TLR7, TLR8 and TLR9, as well as NOD1, which is more often associated with activation by bacterial antigens. Both the SARS-CoV-2 spike protein and adenoviruses such as those used as vectors for spike protein COVID-19 vaccines have innate receptor activation profiles limited to TLR7 and TLR9. Severe COVID-19, in contrast, is characterized by activation (across an average of all such patients) of all the innate receptors studied here, suggesting that non-SARS-CoV-2 antigens are also involved in its pathogenesis. Likewise, each of the autoimmune diseases—MIS-C, Kawasaki disease, autoimmune myocarditis and APS—are similarly characterized by activation of combinations of viral and bacterial receptors, though each in a pattern that may be characteristically different from the others. Since bacteria may provide the additional antigens activating innate immunity in severe cases and autoimmune complications, the typical pattern for Gram-negative bacteria (Figure 3) is provided based on two previous reviews [1,2]. Full references to the relevant primary studies are, once again, provided as a supplement to the References section.

## 3. Receptor Synergisms May Hyper-Activate Innate Immunity

### 3.1. Neutrophil and Monocyte Activation in Severe COVID-19

All innate receptors studied above are characteristic of neutrophils and of monocytes more generally [196], so that the neutrophilia and production of NETs that characterize severe COVID-19 follow naturally from the stimulation of many virus- (especially TLR3, TLR7, NLRP3 and RIG-1) and bacteria-associated (especially TLR2, TLR4, NOD1 and NOD2) innate receptors simultaneously. However, the previous section also demonstrates that SARS-CoV-2 is very unlikely to able to account for this broad stimulation pattern or the consequent cytokine over-production syndrome that characterizes severe COVID-19 and its autoimmune complications because it activates only a few of the virus-associated (endosomal and intracellular) innate receptors (TLR3, TLR7, NLRP3 and RIG-1). SARS-CoV-2 spike-protein-based vaccines activate an even more limited range of innate receptors. Thus, the broader activation profiles found in severe and critical COVID-19 patients, and in patients developing post-COVID-19 autoimmunity, require co-stimulation of bacteria-associated innate receptors. This further stimulation may be due to bacterial (or perhaps fungal) co- or super-infections, for which evidence was presented in the Introduction; additionally, innate stimulation may result from the release of DAMPs, such as cardiolipin, β2GPI, cardiac proteins, etc., from tissues damaged by severe SARS-CoV-2 infections. In this context, it is notable that autoimmune complications associated with COVID-19, such as MIS-C, myocarditis and APS, while differing in the details of the innate receptor activation patterns each displays (Figure 3), occur almost solely within a subset of severe COVID-19 cases. Furthermore, the different DAMPs or the different bacterial or fungal co- or super-infections of SARS-CoV-2 may explain the differences in autoimmune symptoms experienced by COVID-19 patients. Some patients may experience severe COVID-19 without developing autoimmune disease complications, depending on whether they contract any particular co-infection(s) and the degree of cytokine over-production that they experience. *Staphylococci* may synergize with SARS-CoV-2 to produce one set of autoimmune complications, while *Streptococci, Klebsiella* or *E. coli* may produce other sets depending on the mimicry of their antigens for those of their hosts. Autoimmune complications may require specific tissue damage releasing particular autoantigens. Each of these possibilities is discussed in greater depth below.

### 3.2. Synergistic Innate Receptor Activation as a Cause of Cytokine Over-Production and Hyperinflammation in COVID-19

Before exploring how autoimmunity might be induced following COVID-19, it is important to first delve deeper into the etiology of the cytokine over-production and consequent hyperinflammation that accompanies both severe cases and autoimmune complications. What is not evident from the results is that the co-stimulation of extracellular, intracellular and endosomal innate receptors is not merely additive but synergistic. Figure 4 and Figure 5 compare the sets of synergistic interactions triggered by SARS-CoV-2 by itself (Figure 4) and the sets of synergistic interactions resulting from all of the receptors known to be activated in severe and critical COVID-19 (Figure 5). These Figures were generated by transposing the data from Figure 3 (literature review of innate receptor activation) onto the template provided by Figure 2 (summary of known innate receptor synergisms and antagonisms).

While Figure 3 demonstrates that twice as many innate receptors are activated in severe and critical COVID-19 as in an uncomplicated SARS-CoV-2 infection, Figure 4 and Figure 5 demonstrate that five times as many innate receptor synergies are activated in severe and critical COVID-19 than result from uncomplicated SARS-CoV-2. In other words, adding bacterial activation to viral activation results in a far greater cytokine release than is predicted by their simple addition. Synergistic activation of innate immunity provides a reasonable explanation for cytokine over-production in severe and critical cases, as well as a possible model for understanding hyperinflammation accompanying COVID-19 autoimmune syndromes.

Some of the synergisms illustrated in Figure 5 have been studied in detail. For example, bacterial co-infections with SARS-CoV-2 and the presence of bacterial antigens in severe COVID-19 cases have been amply confirmed [20,21,22,23,24]. In particular, Udompornpitak et al. [19] observed a strong correlation between both neutrophilia and cytokinemia in COVID-19 patients and blood concentrations of bacterial lipopolysaccharide (LPS)—a TLR4 activator—as well as with (1→3)-β-D-glucan (BG), which is recognized mainly by dectins on monocytic cells [197]. In the monocyte activation test, BG “powerfully co-stimulated cytokine production (IL-6/IL-8) induced by ligands for TLR1/2, TLR2/6, TLR4, and TLR5” via dectin–TLR synergy [198]. As can be seen in Figure 5, activation of this set of TLRs can be predicted to result in widespread amplification of the synergisms that these receptors have with cytoplasmic and endosomal receptors stimulated by SARS-CoV-2 antigens. As noted above in the results, LPS itself has been shown to be responsible for inducing TLR4-associated activation by the SARS-CoV-2 spike protein [91,92], and the SARS-CoV-2 3a protein can directly activate the NLRP3 inflammasome in LPS-primed macrophages [199]. Thus, bacterial antigens are known to synergize with SARS-CoV-2 and its proteins.

Additionally, SARS-CoV-2 proteins can bind directly to LPS and other bacterial antigens, altering the pathogenicity of both the virus and bacteria. Several studies [200,201,202] have demonstrated that the SARS-CoV-2 spike protein binds with high affinity to LPS and related endotoxins, and that a combination of aerosolized SARS-CoV-2 spike protein with LPS can induce “severe pulmonary inflammation and a cytokine profile similar to that observed in COVID-19” [200]. Conversely, the binding of SARS-CoV-2 to pathogenic bacteria associated with severe COVID-19, such as group A Streptococci and Staphylococcus aureus, has been demonstrated to interfere with bacterial biofilm formation, resulting in the release of more virulent single-celled forms with greater invasive potential, which may lead to severe secondary infections with poor prognosis [203]. In particular, monocytes stimulated via a combination of TLR4, TLR7 and TLR8 by means of a combination of LPS and viral antigens became more permissive to SARS-CoV-2 infection [204], so that a combined SARS-CoV-2-bacterial infection resulted in a positive feedback loop with ever-increased infectivity of both virus and bacterium.

One consequence of the intensive and prolonged increased inflammation experienced by severe and critical COVID-19 patients is the induction of neutrophil and monocyte tolerance to TLR stimulation. Innate receptors on these cells become relatively unresponsive to TLR and NLR agonists ex vivo compared with monocytes from healthy individuals. There are two possible mechanisms for this down-regulation of receptor sensitivity. One is that prolonged or hyper-stimulation of any receptor will cause it to be down-regulated either via second-messenger systems or due to receptor internalization. The other is that some of the synergistic interactions summarized in Figure 2 and Figure 5 can also result in antagonisms (the–and +/− boxes in the Figures), which may become increasingly prominent over time. Several studies have documented that ex vivo neutrophils and monocytes from critically ill COVID-19 patients exhibited impaired cytokine release and consequently a decreased capacity to kill intracellular bacteria via production of reactive oxygen species (ROS) and intracellular myeloperoxidase (MPO) by neutrophils [116,204,205,206]. This decreased ex vivo response to TLR ligand stimulation can be interpreted as immunosuppression [20,207], but may simply be as a natural cellular response to prolonged receptor over-stimulation in vivo, and thus an attempt by the immune system to moderate cytokinemia. Notably, resistance to TLR stimulation disappears during recovery from COVID-19 and is accompanied by the return of normal neutrophil and monocyte function [20,205,206,207,208], as would be expected when the receptors are no longer over-activated.

### 3.3. Innate Receptor Synergies in MIS-C and KD

The innate receptor activation patterns of autoimmune complications associated with COVID-19 mimic the pattern of severe COVID-19 patients as well as symptoms of non-COVID-19 autoimmune diseases [1,2,209]. Because the infections that induced the autoimmune disease have often been cleared several weeks prior, pathogen activation of PAMPs is presumably replaced by ongoing activation by damage-associated self-antigens (DAMPs) that mimic the pathogens, or by circulating immune complexes. Active or ongoing infection by SARS-CoV-2 is not a likely driving force for the autoimmune diseases associated with COVID-19 [210].

A good example of innate receptor synergies in COVID-19 autoimmunity is provided by MIS-C, which generally appears three to six weeks after a SARS-CoV-2 infection has resolved [38,39,40,41] and yet displays an innate activation pattern very similar to that of severe COVID-19 [57,118]. By once again transposing the data from Figure 3 onto the template provided by Figure 2, it is possible to examine the sets of synergistic activation patterns characteristic of MIS-C (Figure 6) and for comparison with KD (Figure 7), uncomplicated SARS-CoV-2 infection (Figure 4) and severe COVID-19 (Figure 5).

MIS-C and KD are each characterized by the activation of nine innate receptors (Figure 3). However, these nine receptors activate sixteen receptor synergisms in MIS-C (Figure 6) and twelve in KD (Figure 7), as compared with six for SARS-CoV-2 (Figure 4) by itself. The increased number of synergisms helps to explain the over-production of cytokines associated with the syndrome [50,51] as compared with an uncomplicated SARS-CoV-2 infection. In MIS-C and KD, as in severe COVID-19, the additional synergisms are due to receptors associated with bacterial antigen activation, particularly TLR2 and TLR4 (Figure 3), and are consistent with high observed rates of bacterial co-infections in severe COVID-19 patients who develop MIS-C (Introduction). This is an important point because some investigators have assumed that MIS-C is caused solely by SARS-CoV-2, which has led them to an unwarranted interpretation of their own data. Wang et al. [57], for example, have documented the activation of the non-canonical inflammasome in MIS-C patients, noting that: “…activation of the non-canonical inflammasome is associated with caspase-4- or caspase-5-dependent pyroptosis, which is known to be induced by intracellular lipopolysaccharide (LPS) from Gram-negative bacteria. Interestingly, the activation of caspase-4/5 also induces the canonical NLRP3 inflammasome…. Because LPS would not be involved in the pathogenesis of MIS-C, [emphasis added] molecular patterns other than those from Gram-negative bacteria must be involved in the induction of non-canonical inflammasome in MIS-C.” In fact, as noted above, the presence of unusually large amounts of LPS and other bacterial antigens have been documented in severe COVID-19 patients with NLRP3 inflammasome activation [19,20,21,22,23,24,197,198,199], and specifically in MIS-C patients [211], so that LPS activation of the non-canonical inflammasome is actually very likely.

The 12 synergies predicted for both the MIS-C and KD activation patterns are each distinct subsets of the 29 possible synergisms that can develop in severe COVID-19, suggesting that autoimmune sequelae following COVID-19 (or, in the case of KD, unknown infections) involve ongoing activation of select groups of the innate receptors that can be activated in COVID-19. Since the sets of receptor synergisms activated in MIS-C are different than those activated in KD, different infectious triggers are likely involved in their pathogenesis. Reference to Figure 3 suggests that some of the DAMPs driving ongoing autoimmunity in both diseases may be myosin and cardiolipin (CL), both of which are cardiac antigens that mimic antigens found in bacterial co-infections associated with severe COVID-19 [39,150,153]. Anti-CL antibodies have also been documented in some MIS-C patients [212,213]. These two DAMPs do not, however, account for the range of innate receptors activated in either disease, and despite the fact that both MIS-C and KD are characterized by vasculitis, anti-neutrophil cytoplasmic antibodies (ANCAs), which characterize other forms of autoimmune vasculitis [214], including those sometimes associated with COVID-19 infections [215,216], have not been reported in either MIS-C or KD [212]. Other possible autoimmunogens, such as C-reactive protein (CRP), alpha2 macroglobulin (α2M) and serum amyloid P (SAP) have been documented in MIS-C [211], warranting additional investigation.

### 3.4. Innate Receptor Synergisms in APS

Anti-CL antibodies are also very common among people who develop anti-phospholipid syndrome (APS). In APS, autoimmunity is directed against phospholipids and their binding proteins, resulting in coagulopathies, whether associated with COVID-19 [217] or not [218]. However, other autoantibodies against blood proteins are also found in severe COVID-19 cases. In one study of 54 COVID-19-hospitalized patients (34 in the intensive care unit (ICU) and 20 in non-ICU), 74.1% were positive for aPL antibodies, with 60% testing positive for lupus anti-coagulant (La), 18.5 % for IgM aCL, 14.8% for IgM anti-β2GPI and 24% for IgA anti-β2GPI [219]. Individually, these antibodies did not predict thrombotic events, which occurred in only nine patients (see also [70]), but combinations of these antibodies were predictive, suggesting that multiple autoantigens need to be targeted to result in actual autoimmune disease manifestations [84,85]. Thus, COVID-19 coagulopathies are characterized by the presence of multiple autoantibodies besides anti-CL that include anti-β2GPI, anti-platelet factor 4 (PF4), anti-phosphatidylserine/prothrombin (aPS/PT), and anti-lupus coagulant (anti-La antigens), which are not usually present in MIS-C or KD. In consequence, the innate receptor activation pattern associated with APS (Figure 8) differs, once again, from an uncomplicated SARS-CoV-2 infection (Figure 4), severe COVID-19 (Figure 5), MIS-C (Figure 6) and KD (Figure 7), suggesting a different set of co-infections may be involved in its pathogenesis. Since SARS-CoV-2 is the common denominator in all COVID-19-associated autoimmune diseases, it is logical to assume that what differentiates each of these autoimmune sequelae is a different set of bacterial (or fungal) co- or super-infections. Alternatively, perhaps SARS-CoV-2 targets varied tissues in patients as a result of different underlying pre-existing conditions that release DAMPs that initiate specific autoimmune reactions. While the existing data does not permit an analysis of which co-infections may be involved in any given COVID-19 autoimmune disease (the published literature does not specify which particular bacteria are responsible for increases in LPS and other bacterial antigens in the blood of patients), there is evidence in APS that bacterial antigens are very likely to be involved in driving the increased inflammation that is required to support the autoimmune disease process [219]. Once again, there is direct evidence for LPS stimulation of TLR4 as an activator of the inflammasome in APS, driving thrombosis [220,221,222]. LPS may be replaced in the ongoing disease process by the DAMP heat shock protein 60 (HSP60) [220], so that once tissue damage has been initiated, it may continue to drive pathology. LPS further synergizes with both aPS/PT and aβ2GPI IgG antibodies to drive TLR activation [223,224]. Additionally, endotoxemia more generally is itself associated with thrombotic risk, independent of COVID-19 [225], and one of the functions of β2GPI is to bind directly to bacterial cell membranes to destabilize them, potentially causing cell death [226]. In fact, β2GPI binds directly to LPS [227,228] demonstrating the molecular and antigenic complementarity of the two molecules. Moreover, various microbial proteins from bacteria such as *Haemophilus influenzae* [[17],[228],[229],[230],], *Escherichia coli* [229,230,231] and *Staphylococcus aureus* [229,230,231], and viruses such as cytomegalovirus and tetanus toxoid (TTd) [228], mimic β2GPI, so that a standard animal model of APS has been developed by combining either LPS or one of its standard bacterial sources, *M. tuberculosis*, with TTd [232,233]. These observations help to explain the clear evidence that multiple TLR and NLR synergisms involving the activation of both “viral” and “bacterial” receptors are evident in producing the hyperinflammatory state typifying APS among COVID-19 patients (Figure 8).

Once again, as in severe COVID-19 and MIS-C, SARS-CoV-2 by itself (Figure 4) cannot account for the set of TLR and NLR synergisms observed in APS and summarized in Figure 8. In this case, recent experiments have demonstrated that antibodies against SARS-CoV-2 proteins recognize only some of the main autoantigens associated with COVID-19 coagulopathies, specifically the prothrombin component of aPS/PT, Factor VIII and von Willebrand Factor [230,231] (Table 3). Multiple studies have concluded that antibodies against SARS-CoV-2 or its proteins do not induce autoantibodies against CL [230,231,234,235] or β2GPI [230,231,234,235], and the consensus is that it does not cross-react with PF4 either [76,77,78,230,231,236,237]. Thus, additional microbial antigens mimicking these targets, or release of the autoantigens themselves as DAMPs, are required.

Bacteria associated with severe COVID-19 are good candidates, since they express CL in their cell membranes [162,163], and cross-reactivity between some of these bacterial antibodies and both β2GPI [17,230,231] and PF4 has been demonstrated experimentally [231]. In sum, the autoantigens targeted in APS can only be accounted for by combining a SARS-CoV-2 infection with particular bacterial co- or super-infections or the presence of autoantigenic DAMPs. Such combinations are also necessary to activate the synergistic sets of TLR–NLR receptors required to produce the increased inflammation supporting the induction of subsequent autoimmune disease (Figure 8).

### 3.5. SARS-CoV-2 Vaccines and Risks of Autoimmune Complications

The preceding data lead logically to the question of the safety of SARS-CoV-2 vaccines, which, not incidentally, provide a real-world test of many of the conclusions reached above. On the one hand, unadjuvanted whole-virus SARS-CoV-2 vaccines, such as the Sinovac-CronaVac vaccine, should result in no more innate stimulation than the virus itself, as summarized in Figure 3 and Figure 4. Thus, the risk of autoimmune complications and increased innate activation, *in the absence of a bacterial co-infection or release of DAMPs from damaged tissue,* should be no greater than that observed in mild or symptom-free SARS-CoV-2 infections. Adenovirus-vectored SARS-CoV-2 spike protein vaccines, such as ChAdOx1 nCoV-19 (the AstraZeneca adenovirus 5-based vaccine) in healthy individuals should be even safer from an innate activation perspective. Although both the spike protein and the adenovirus component independently activate some TLR and NLR (Figure 3), the number of innate receptors activated is significantly less than in the case of whole SARS-CoV-2 virus, and the protein–adenovirus combination induces no known innate receptor synergisms and probably one antagonism (Figure 9). If increased inflammation is required to support autoimmune complications, these vaccines, on their own, lack the ability to induce such complications. Evaluating the innate immune activation resulting from SARS-CoV-2 spike protein vaccines, such as BNT162β (the Pfizer–Biontech mRNA vaccine), is more difficult to predict in the absence of formal studies of innate receptor activation by the nanoparticles. The only relevant study suggests that such nanoparticles may not activate, or may even down-regulate, innate receptors [238], and so would not contribute to inflammatory risk. Thus, nanoparticle-delivered SARS-CoV-2 mRNA vaccines delivered to healthy individuals should be particularly safe in terms of innate activation.

In fact, SARS-CoV-2 vaccines have proven to be extremely safe in general, and studies looking for the induction of autoantibodies have generally failed to do so. For example, several investigators have documented mimicry of PF4 for SARS-CoV-2 proteins [230,231,234,235,237] so that it could be predicted that COVID-19 vaccines might induce autoantibodies that cross-react with PF4. In fact, depending on the cut-off value used for significance, transient anti-PF4 antibodies were found in 5.6% of BNT162β (the Pfizer–Biontech mRNA vaccine)- and between 8.0 and 67% of ChAdOx1 nCoV-19 (the AstraZeneca adenovirus-5-based vaccine)-inoculated people [71,239,240,241]. However, none of the thousands of patients in these studies developed clinically overt coagulopathies. Thus, measurement of antibodies against individual autoantigens are of limited value in predicting complications [77,240,241,242,243]. This conclusion has also been reached by studies investigating increased incidence of CL antibodies following SARS-CoV-2 vaccination [71,72,244,245,246,247]. Only one such study found any evidence of increased titers of such antibodies—in this case linked to the ChAdOx1 nCoV-19 vaccine—but concluded that these were transient and of no clinical significance [243]. These results are consistent with the fact that SARS-CoV-2 proteins do not mimic CL and are unable to induce cross-reactive antibodies [230,231,234,235,237]. A number of studies have documented a similar lack of increased antibody titers against CL, β2GP1, PF4-heparin and phosphatidylserine/prothrombin (aPS/PT) following use of the Sinovac–CoronaVac inactivated whole-SARS-CoV-2-virus vaccine [72,244,248], with only seven cases developing low-titer anti-PF4-heparin antibodies (range: 1.18–1.79 U/mL) after vaccination, and none of these exhibiting any sign of thrombotic disorder [72]. Thus, if the degree of innate immune system activation is very limited, as is the case with SARS-CoV-2 spike protein or even whole-virus vaccines (Figure 3 and Figure 4) and there are no receptor synergisms (as in Figure 9) to drive cytokine over-production, then no autoimmune disease results even in the presence of autoantibodies. Therefore, under normal circumstances, these vaccines are very safe from an autoimmune complication perspective.

So, what may cause the very rare coagulopathies, myopathies and other autoimmune sequelae that sometimes follow SARS-CoV-2 vaccination? As with the autoimmune complications associated with the viral infection, a likely contributor to post-vaccinal complications are bacterial (and possibly fungal) co-infections. Here, it is important to reiterate the point that while individual autoantibodies do not predict the incidence of COVID-19 coagulopathies and combinations of anti-CL and anti-β2GPI antibodies, as well as, in some cases, antibodies against additional blood proteins such as collagens, von Willebrand Factor and prothrombin [70,84,85,216,217,218,219], do. As Table 3 illustrates, this combination of autoantibodies cannot be induced by SARS-CoV-2 by itself but requires bacterial antigens to be present as well. Assume, therefore, that a patient presents with a (perhaps undiagnosed) bacterial infection at the time they are vaccinated with a SARS-CoV-2 vaccine, or that they are exposed immediately afterwards to such an infection. As Figure 10 illustrates, combining a bacterial infection with a SARS-CoV-2 spike protein vaccine may result in nineteen synergistic receptor interactions where, in the absence of a bacterial infection, there were none (Figure 9). The current literature strongly suggests that adenovirus-vectored vaccines are more likely to be associated with post-vaccinal autoimmune complications than are nanoparticle vaccines [71,239,240,241,242,243,244,245,246,249]. This phenomenon is also predicted in Figure 10 and may have its basis in the additional four receptor synergisms that the adenoviral vector adds to the activation pattern. It can be predicted that whole-virus vaccines, such as the Sinovac–CoronaVac vaccine, have an even greater probability of inducing complications than either the nanoparticle or adenovirus-vectored SARS-CoV-2 spike protein vaccines because, in the context of a coexisting bacterial infection, the resulting synergistic activation pattern will resemble that of the severe COVID-19 patients illustrated in Figure 5. No clinical studies appear to address this hypothesis in the specific context of COVID-19 vaccinations, and therefore it must stand as a testable prediction of the analysis presented here.

However, it must be stressed that some of the increased complication risks and predicted receptor synergies shown in Figure 10 have been demonstrated in various types of clinical, in vivo and in vitro studies. Adenoviruses and adenovirus vectors have previously been associated with increased risk of coagulopathies [250,251,252,253,254,255] and are known to synergize with bacterial antigens such as LPS to produce increased inflammatory responses [111,256,257]. *Streptococcal* and *Staphylococcal* infections are also independently associated with increased risks of both coagulopathies [258,259,260,261] and cardiomyopathies [262,263,264], and virus–bacteria co-infections often occur together in patients who subsequently develop these autoimmune diseases (reviewed in [59,60,61,62]). Combined SARS-CoV-2–bacterial infections might therefore explain both the observed association between the severity of COVID-19 and increased risk of cardiac complications [265], as well as the rarity of vaccine-associated cardiac complications among presumably healthy vaccine recipients.

The type of innate receptor activation pattern analysis performed here may also have applications in understanding and predicting the utility and risks of adding specific innate receptor agonists or antagonists to SARS-CoV-2 vaccines in order to improve their immunogenicity and lower their risk of complications. For example, adjuvanting SARS-CoV-2 vaccines with TLR4, TLR9, RIG-1, etc. agonists improves the ability of the vaccines to induce neutralizing antibodies [266,267,268,269]. However, in the context of calculating risks of subsequent autoimmune complications of such adjuvanted vaccines in the presence of possible bacterial or fungal co-infections, some of these adjuvants may pose greater or lesser risks. Such considerations may help vaccine developers improve immunogenicity in ways that also minimize hyperinflammatory risks and vaccine complications.

### 3.6. Possible Roles of Underlying Diseases That Predispose Severe COVID-19

One final consideration must also be taken into account in evaluating the hyperinflammatory and autoimmune disease risks of people exposed to SARS-CoV-2 and its vaccines, and that is the role of underlying diseases such as diabetes that predispose severe COVID-19. A complete analysis of how these risks are related to the activation of innate immunity in advance of infection or vaccination would require another paper of the same length as this one, so suffice it to say here that abundant evidence exists for the increased activation of discrete sets of innate receptors in both type 1 diabetes (TLR2, TLR4, TLR7/8 and TLR9 [269]) and type 2 diabetes (TLR2 and TLR4 [270]) compared with healthy control groups. Starting with a higher degree of innate activation than normal may result in a greater response to COVID-19-associated infections and thus increase the risk of autoimmune complications. Disrupted gut microbiomes may be partially responsible for this innate stimulation, particularly by causing leaky gut syndrome, permitting increased amounts of LPS to enter into the blood stream [271] and setting the stage for the types of virus–bacterium synergies discussed here. In short, as complex as the analysis performed here may seem, it only scratches the surface of the actual complexity that exists in real-world COVID-19 cases.

## 4. Future Directions for Further Research and Implications for Prevention and Treatment of COVID-19-Associated Autoimmune Diseases

The take-home message of this paper is that the risk of autoimmune diseases complicating COVID-19 appears to be highly associated with the severity of COVID-19, and both correlate highly with the neutrophilia and production of NETs that result from innate receptor patterns, and that can best be explained by the synergistic activation of receptors generally associated with combinations of viral and bacterial antigens. The reasoning is that if SARS-CoV-2 only activates TLR 3,7,9, RIG-1 and MDA5 and no NOD, but severe SARS-CoV-2 is characterized by the activation of TLR 2,3,4,7,9, RIG-1, MDA5 and NOD2, then we must account for the activation of TLR2, TLR4 and NOD2 through non-SARS-CoV-2 antigens. This reasoning does not say anything about what is “required” for disease to be manifested, but is rather about accounting for observed innate activation patterns in disease manifestations. While one possible etiology for the innate receptors activated in severe COVID-19 and its autoimmune complications involves bacterial antigens, another possibility is activation by self-antigens released as a result of SARS-CoV-2-induced cellular destruction. An additional possibility is that bacteria mimicking self-antigens induce the autoimmune process, which is then supported by these self-antigens after the bacterial infection is cleared. Because the innate activation patterns differ from one autoimmune disease to another, it is logical to assume that different (sets of) bacteria (or perhaps fungi [272,273]) and/or DAMPs are associated with risks of different autoimmune diseases, thereby explaining how SARS-CoV-2 can be associated with multiple autoimmune disease sequelae. Adenovirus vectors and/or co-infections may also increase the risk of developing autoimmune complications by activating additional synergistic innate receptor combinations beyond SARS-CoV-2 or its proteins (Figure 11).

As noted repeatedly throughout the Introduction, TLRs and NLRs can be activated by DAMPs as well as PAMPs, so the role of DAMPs in supporting COVID-19-associated autoimmunity must also be considered. Unfortunately, much less is known about DAMP activation of innate immunity than is known about PAMP activation, so concrete conclusions are difficult to make. Figure 12 summarizes several reviews concerning DAMP activation of TLRs and NLRs [1,274,275,276,277]. The overall picture is that DAMPs, as a general class, may activate all innate receptors. Mitochondrial antigens that might be released during any type of cellular or tissue damage are known, for example, to activate NLRP3, TLR9, RIG-1 and MDA-5 [275,276,277], and, in particular, to act like bacterial mitochondrial antigens [278]. However, particular DAMPs associated with damage to specific tissues in COVID-19 autoimmune diseases, such as myosin, collagens or fibrinogen, have TLR activation profiles limited to TLR2 or TLR4 [1,275]. Apparently, none of these antigens have been tested to determine the degree to which they do or do not synergize with antigens expressed by SARS-CoV-2 or other viruses. Thus, it is not known whether DAMPs are involved in triggering autoimmune diseases due to tissue or cellular damage caused by microbial infections, or whether DAMPs replace some of the microbial antigens to support innate immune stimulation after microbes trigger autoimmunity that persists after the microbial infection is resolved. It seems unlikely that DAMPs *by themselves* can trigger autoimmune disease for the simple reason that any type of tissue damage releases DAMPs that activate innate immunity, but evidence linking tissue damage per se to the induction of autoimmune disease is lacking. Clearly, more research into the various roles played by PAMPs, DAMPs and their interactions in autoimmune diseases is much needed.

The evidence connecting the development of COVID-19 autoimmune complications to bacterial (and possibly fungal) co-infections of SARS-CoV-2 patients complicates the issue of antibiotic stewardship in the treatment of COVID-19 (e.g., [279,280,281]). However, if the difference between severe COVID-19 and mild or moderate cases is whether bacterial (or perhaps fungal) co-infections are present, then some means of protecting SARS-CoV-2-infected people from such co-infections is needed not only to prevent admissions to intensive care units but also to decrease or eliminate the potential for autoimmune sequelae. The analysis presented here makes it imperative that protection be implemented by some means, whether it is by prophylactic use of antibiotics or by other measures.

One alternative approach to protecting people against COVID-19 autoimmune complications that preserves antibiotic conservation might be to expand vaccination against bacterial co-infections associated with severe COVID-19, such as *Haemophilus influenza* (Hib vaccine) and group A *Streptococci* (pneumococcal vaccines). Notably, multiple studies examining hundreds of thousands of patients have documented some protection afforded by Hib vaccination, and a very significant protection against COVID-19 morbidity and mortality from pneumococcal vaccination [282,283,284,285,286,287,288,289,290,291]. No studies seem to exist as to whether this protection extends to COVID-19-associated autoimmune complications, but such studies may be warranted. It must, however, be borne in mind that Hib and Streptococcal infections are only two among many types of bacterial infections complicating moderate and severe COVID-19 cases, and therefore only some types of complications may be moderated. Additionally, since tetanus toxin mimics β2GPI [228,232,233], tetanus-antigen-containing vaccines may also moderate the risk of some types of COVID-19 coagulopathies.

Another approach to prophylaxis against bacterial and fungal co-infections of SARS-CoV-2 might be increased healthcare provider and patient awareness of the autoimmune disease risks that may attend such co-infections. If the analysis carried out here is accurate, then a simple SARS-CoV-2 infection that may present with mild or moderate symptoms can be transformed into a severe case with very significantly increased risks of autoimmune complications by acquiring a co- or super-infection. Advocacy for isolating, using personal protective equipment, hand-washing, proper oral hygiene, etc., following a positive COVID-19 test may significantly reduce such risks in healthcare, business and home environments.

Similar practices may reduce the risk of post-vaccination autoimmune complications. People might be screened for evidence of infections (e.g., presence of diarrhea, gum disease, infected wounds, fever, etc.) prior to SARS-CoV-2 vaccination, and vaccination delayed until the infections are treated or resolved. Equally importantly, newly vaccinated people could be advised to use precautions to prevent exposure to infection for a reasonable period of time (perhaps a week or ten days) following their vaccination.

There is a clear need to understand the role of innate activation patterns that pre-exist among individuals at high risk for COVID-19 severity and complications. Dysbiosis, pre-existing active autoimmune disease, chronic infections, heart or kidney damage and so forth may all create inflammatory conditions that can release DAMPs and synergize with SARS-CoV-2 to produce a hyperinflammatory environment conducive to subsequent autoimmune disease.

New types of experiments are called for. While many combinations of innate receptor antagonists have been explored (see the Supplement to References), very little is known about how SARS-CoV-2 interacts with the bacterial, fungal and viral co- and super-infections that characterize severe COVID-19. SARS-CoV-2-susceptible animals, such as golden hamsters, might be infected with combinations of the virus with Streptococci, Staphylococci, E. coli, Klebsiella, etc., to explore how these microbes interact and whether they induce autoimmune complications similar to those reviewed here. Similarly, it would be interesting to know whether the course of COVID-19 differs in animals with diabetes, heart disease or dysbiosis which are subsequently exposed to SARS-CoV-2.

New types of clinical studies are also needed. The data currently available do not permit analysis of whether particular co- or super-infections predispose any particular autoimmune complication. For example, a recent paper has reported evidence from T-cell studies that MIS-C patients may be unusually likely to have been infected with Enterococcus faecium. There appear to be no studies of rates of E. faecium infections among children who develop MIS-C to confirm or invalidate the link. Similarly, Streptococcal infections have been associated with autoimmune myocarditis prior to COVID-19, but this association has not been tested for COVID-19; nor have the associations of Staphylococcal and adenovirus infections with coagulopathies that preceded COVID-19 been examined in regard to the current pandemic. Working out the etiologies of any of these autoimmune complications would provide insights not only into the mechanisms of these human autoimmune diseases but also provide clues to their prevention and treatment.

Finally, combinations of innate receptor antagonists might be very effective in preventing or moderating COVID-19-associated autoimmunity by moderating the underlying proinflammatory environment. Since both TRIF and MyD88 pathways are activated in severe COVID-19 and its autoimmune disease complications, and both the typical and atypical inflammasome pathways, either broadly acting inhibitors, or combinations of several specific inhibitors, will probably be needed. Broadly acting inhibitors that work simultaneously against multiple innate receptors that have proven to have efficacy in decreasing the risk of severe COVID-19 include melatonin (reviewed in [2]) and steroids [292,293]. Whether such interventions specifically reduce the risk of autoimmune complications as well has not yet been investigated. One might predict from Figure 2 and Figure 5 through 8 that a minimum of a combination of individual small-molecule or monoclonal antibody antagonists against TLR3, TLR4 and NOD1 or NOD2 might be required to produce protection similar to, or better than, these natural compounds.

## 5. Materials and Methods

An intensive effort was made to identify all studies available on PubMed and MedLine of innate immune activation of TLR, NLR, RIG-I and MDA-5 related to the SARS-CoV-2 virus, the SARS-CoV-2 spike protein, adenovirus and adenovirus vaccine vectors, severe COVID-19, MIS-C, KD, anti-phospholipid syndrome and specific antigens or antibodies associated with these (e.g., anti-SARS-CoV-2 antibodies, anti-cardiolipin (aCL) antibodies, anti- β2GPI (aβ2GPI) antibodies, anti-lupus antigen (aLa), anti-myosin antibodies, etc.). The search was limited to human clinical and ex vivo studies to ensure applicability of the results to understanding human disease, except where no human data were currently available. Where data were lacking or insufficient, an additional search was made of Google Scholar sources that, in some cases, revealed papers published in journals not listed in the National Library of Medicine catalogue.

TLR, NLR, RIG-I and MDA-5 activation or antagonism were selected for analysis because of the abundance of available research on the activation of these particular receptors in COVID-19 and its associated autoimmune diseases. As noted in the Introduction, C-type lectin receptors (CLR) were not investigated because they consist of a very diverse group of receptors that respond to both viruses and bacteria, making prediction of their effects in any given disease difficult and also because evidence regarding their activation of antagonism in COVID-19-related diseases is much sparser than for the other classes of innate receptors. Further research into CLR might therefore modify some of the details of the analysis that follows.

## Figures and Tables

**Figure 1 ijms-24-03001-f001:**
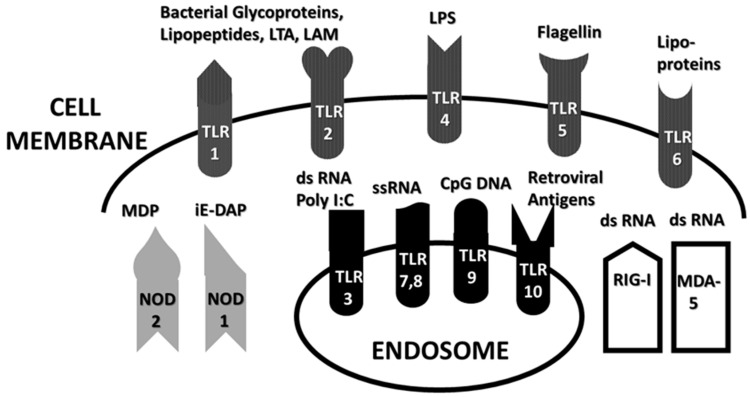
Schematic diagram summarizing locations (cell membrane, cytoplasm or endosome) on human innate immune system cells of the Toll-like receptors (TLR), nucleotide-binding oligomerization domain-containing proteins (NOD), retinoic acid-inducible gene I (RIG-1)-like receptors (RLR), RIG-I and the anti-melanoma differentiation-associated gene 5 (MDA-5). The main activating ligands are also shown: MDP-N-acetyl-muramyl-D-alanyl-isoglutamine-containing peptides; iE-DAP-γ-D-glu-meso-diaminopimelic acid (iE-DAP) dipeptide-containing antigens; LPS—lipopolysaccharides; poly I:C—a polymer of inosine and cytosine that mimics double-stranded RNA (dsRNA); ssRNA—single-stranded polyribonucleic acids; CpG DNA—cytosyl-p-guanosyl oligodeoxynucleotide deoxyribonucleic acid; dsDNA—double-stranded DNA; LTA—Lipoteichoic Acid; LAM—Lipoarabinomannan. LTA and LAM are found only in certain classes of bacteria.

**Figure 2 ijms-24-03001-f002:**
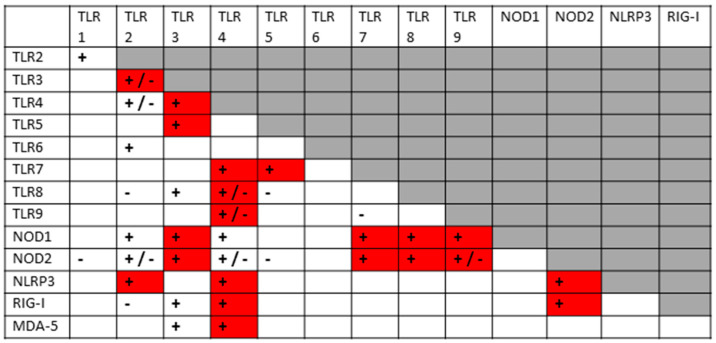
Summary of currently known synergisms (+), antagonisms (−) and cases where pretreatment (usually several days or more in advance of the second stimulus) by one receptor agonist results in antagonism to the other, whereas co-stimulation results in synergism (+/−). Synergies involving virus-activated receptors (TLR3, TLR7, TLR8, TLR9, RIG-I or MDA-5) with bacteria-activated receptors (TLR2, TLR4, TLR5, NOD1 or NOD2) are highlighted in red because these are of particular importance for understanding how co-infections between SARS-CoV-2 and bacteria may result in increased inflammation. Blank boxes indicate that activators of the pair of TLRs or NLRs do not synergize, or that no studies of their possible synergy could be found. The gray boxes are blocked out because they duplicate the red and white boxes. The Figure is based on an integration of hundreds of studies provided in two recent reviews [1,2], supplemented with several additional sources [5,6,7,8,9,10,11,12]. The full set of original references is appended as a Supplementary References in Appendix A.

**Figure 3 ijms-24-03001-f003:**
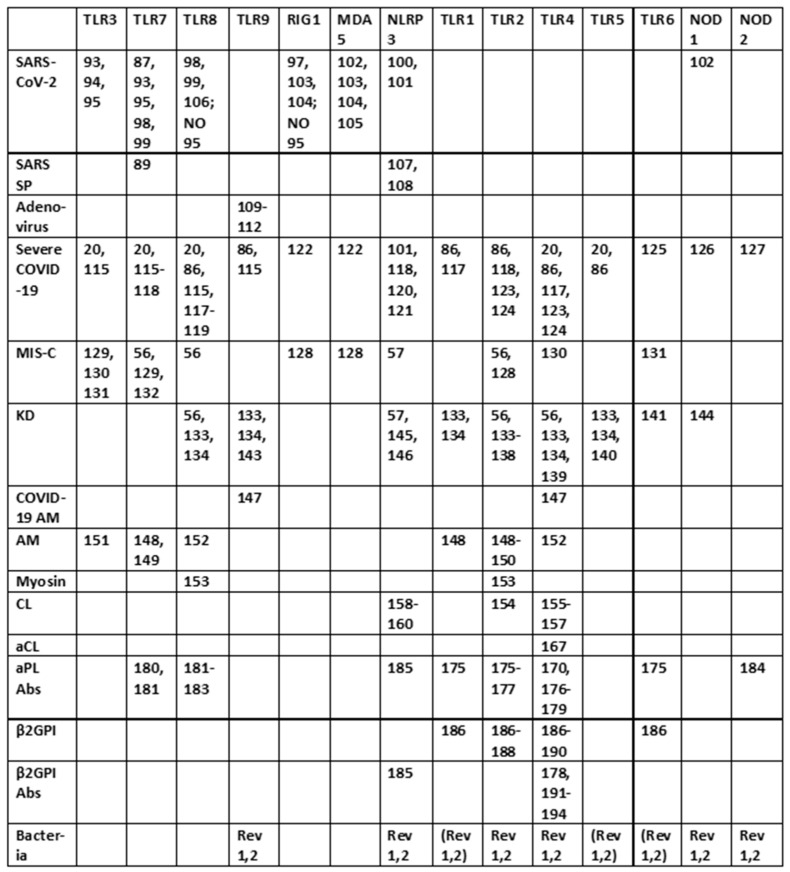
Summary of results. Each number refers to an article cited in the References. Rev = “reviewed in”; (Rev 1,2) indicates that these TLR activations only occur for some bacteria. Full references to the original literature on bacterial activation of innate receptors informing [1,2] are provided as a Supplement to the References. Blank entries indicate in most cases that studies found no cross-reactivity between human antibodies or sera with the corresponding antigen; in a few cases, such cross-reactivity has not yet been tested. SARS SP = SARS-CoV2 spike protein; MIS-C = Multisystem Inflammatory Syndrome in Children, sometimes called Pediatric Inflammatory Multisystem Syndrome Temporally Associated with SARS-CoV-2; AM = autoimmune myocarditis; CL = cardiolipin; aCL = anti-cardiolipin antibody; aPL = anti-phospholipid; Abs = antibodies; β2GPI = beta 2 glycoprotein I; see Figure 1 for TLR, NOD, NLRP, RIG-I and MDA-5.

**Figure 4 ijms-24-03001-f004:**
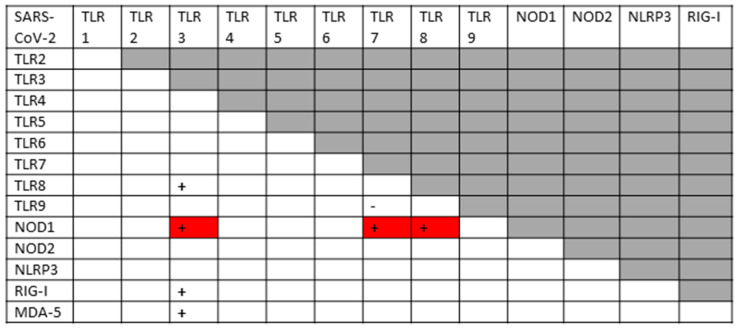
Predicted synergisms (+) and antagonisms (−) resulting from SARS-CoV-2 activation of innate receptors resulting from mapping Figure 3 (literature review of innate receptor activation) onto the template provided by Figure 2 (summary of known innate receptor synergisms and antagonisms). Red entries indicate extracellular (“bacterial antigen”) receptor synergisms with intracellular and endosomal (“viral antigen”) receptors.

**Figure 5 ijms-24-03001-f005:**
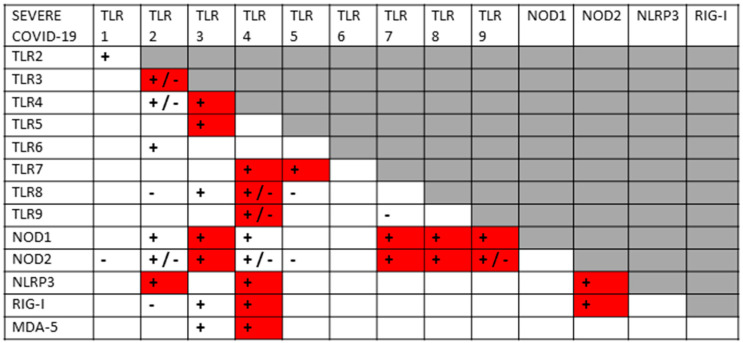
Predicted synergisms (+) and antagonisms (−) resulting from severe COVID-19 activation of innate receptors resulting from mapping Figure 3 (literature review of innate receptor activation) onto the template provided by Figure 2 (summary of known innate receptor synergisms and antagonisms). Red entries indicate extracellular (“bacterial antigen”) receptor synergisms with intracellular and endosomal (“viral antigen”) receptors.

**Figure 6 ijms-24-03001-f006:**
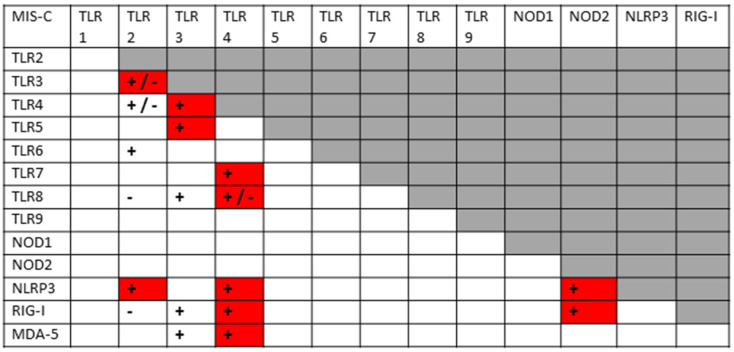
Predicted synergisms (+) and antagonisms (−) resulting from multisystem inflammatory syndrome in children (MIS-C) activation of innate receptors resulting from mapping Figure 3 (literature review of innate receptor activation) onto the template provided by Figure 2 (summary of known innate receptor synergisms and antagonisms). Red entries indicate extracellular (“bacterial antigen”) receptor synergisms with intracellular and endosomal (“viral antigen”) receptors.

**Figure 7 ijms-24-03001-f007:**
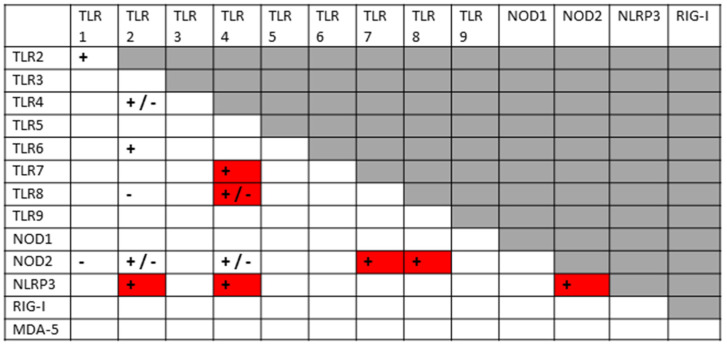
Predicted synergisms (+) and antagonisms (−) resulting from Kawasaki disease activation of innate receptors resulting from mapping Figure 3 (literature review of innate receptor activation) onto the template provided by Figure 2 (summary of known innate receptor synergisms and antagonisms). Red entries indicate extracellular (“bacterial antigen”) receptor synergisms with intracellular and endosomal (“viral antigen”) receptors.

**Figure 8 ijms-24-03001-f008:**
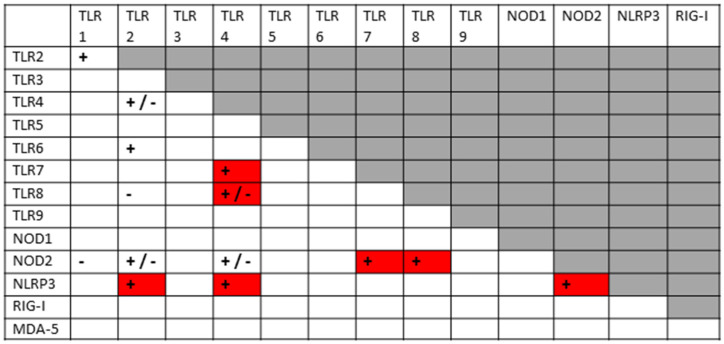
Predicted synergisms (+) and antagonisms (−) resulting from anti-phospholipid syndrome activation of innate receptors resulting from mapping Figure 3 (literature review of innate receptor activation) onto the template provided by Figure 2 (summary of known innate receptor synergisms and antagonisms). Red entries indicate extracellular (“bacterial antigen”) receptor synergisms with intracellular and endosomal (“viral antigen”) receptors.

**Figure 9 ijms-24-03001-f009:**
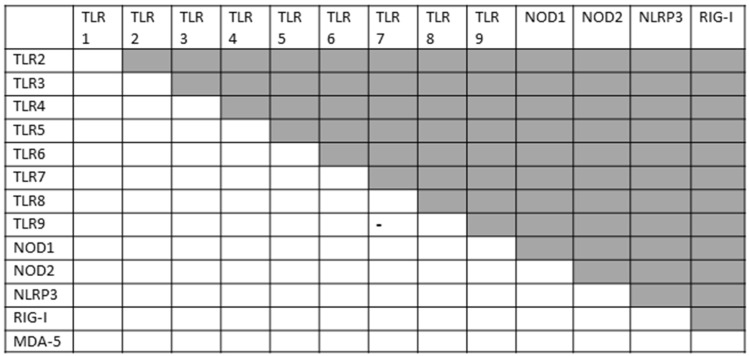
Predicted synergisms (+) and antagonisms (−) resulting from adenovirus-vectored SARS-CoV-2 spike protein vaccine activation of innate receptors resulting from mapping Figure 3 (literature review of innate receptor activation) onto the template provided by Figure 2 (summary of known innate receptor synergisms and antagonisms).

**Figure 10 ijms-24-03001-f010:**
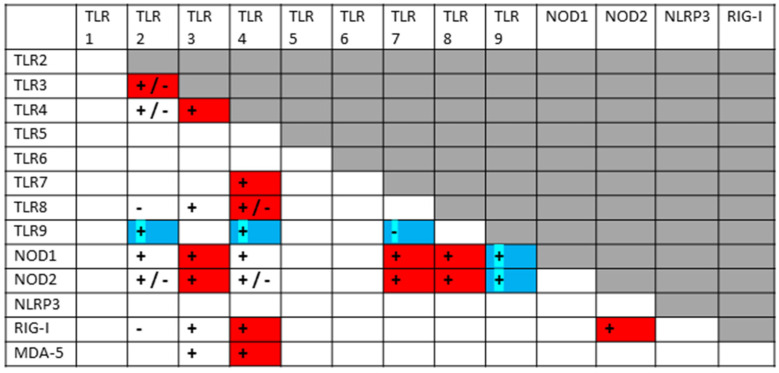
Predicted synergisms (+) and antagonisms (−) resulting from SARS-CoV-2 spike protein vaccine activation of innate receptors in the presence of a bacterial infection such as Staphylococcus or Streptococcus. The blue entries indicate additional synergisms and antagonisms that would result if the spike protein vaccine were delivered using an adenovirus vector. Interactions were predicted by mapping Figure 3 (literature review of innate receptor activation) onto the template provided by Figure 2 (summary of known innate receptor synergisms and antagonisms). Red entries indicate extracellular (“bacterial antigen”) receptor synergisms with intracellular and endosomal (“viral antigen”) receptors.

**Figure 11 ijms-24-03001-f011:**
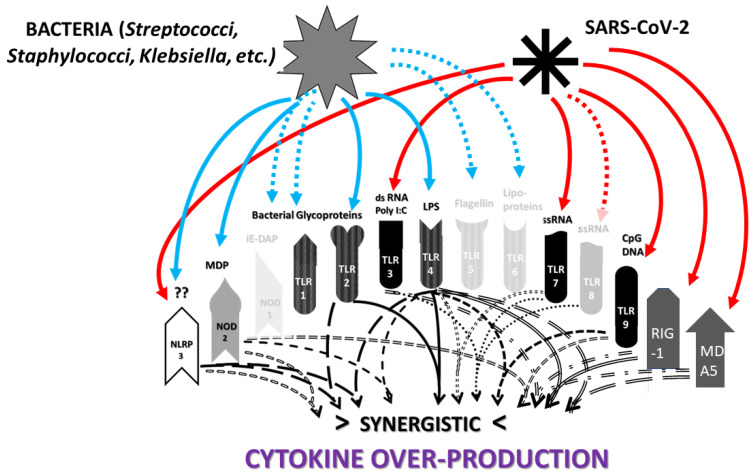
Summary of innate receptor activations produced by SARS-CoV-2 (red lines) and bacteria (blue lines) and their synergisms. Solid red and blue lines indicate activation of receptors that has been well-validated by experimental or clinical studies and that, in the case of bacteria, are common to all bacterial co- and super-infections commonly observed in severe COVID-19. Dotted red lines indicate possible activation by SARS-CoV-2 antigens that are not well-established. Dotted blue lines indicate receptor activation by only some species of bacteria that co- or super-infect SARS-CoV-2 in severe COVID-19. The black lines indicate pairs of receptors involving well-established activation patterns that produce synergisms. Receptors that may be activated (dotted red and blue lines) are not included in these synergisms, but may produce additional synergisms that add to the cytokine over-production. Note that TLR4 (bacteria-activated through LPS) is particularly prolific in its range of synergisms with SARS-CoV-2-activated receptors.

**Figure 12 ijms-24-03001-f012:**
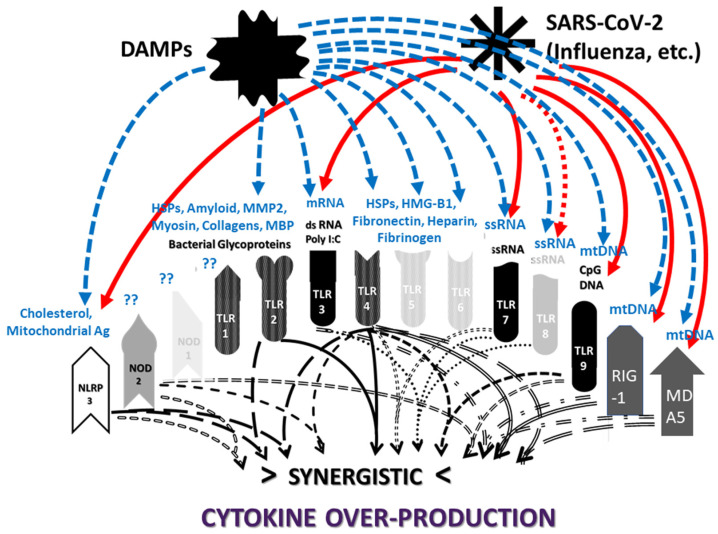
Summary of innate receptor activations produced by SARS-CoV-2 (red lines), damage-associated molecular pattern (DAMP) antigens (blue lines) and their synergisms. Solid red lines indicate activation of receptors that have been well-validated by experimental or clinical studies. Dotted red lines indicate possible activation by SARS-CoV-2 antigens that are not well-established. Dotted blue lines indicate receptor activation by DAMPs that may or may not be released in any given patient with severe COVID-19 and may therefore be disease-, organ-, or tissue-specific. The black lines indicate pairs of receptors involving well-established activation patterns that produce synergism in COVID-19 (see Figure 11). Ag = antigens; ?? indicates that DAMP activators are not known; HSP = heat-shock proteins; MMP2 = matrix metalloproteinase-2 (collagenase); MBP = myelin basic protein; mRNA = messenger ribonucleic acid; HMG-B1 = High-mobility group box 1 protein, ssRNA = single-stranded ribonucleic acid; mt DNA = mitochondrial deoxyribonucleic acid.

**Table 1 ijms-24-03001-t001:** Comparison of various immunological markers in MIS-C patients with mild-to-moderate COVID-19 and severe COVID-19 patients, demonstrating that MIS-C patients have even greater increased inflammation than the average severe COVID-19 patient. * The Son et al. review [51] included multiple reviews that appear to have used some overlapping data sets; the largest reported 1116 patients; if there was no overlap, the total of all the studies would approach 4000 cases. ^ Garcia-Salido et al. [63] reported that 84% of MIS-C cases had septic shock. # Davies et al. [64] (*n* = 78 patients) reported that up to 90% of their British PIMS cases experienced septic shock. ~ Quereshi et al. [65] (*n* = 8163 patients) found a higher rate of stroke than did Kornitzer et al. [48]. + Smilowitz et al. [66]. = Fu et al. [67] (*n* = 75 patients). KD = Kawasaki disease.

	MIS-C [51] (*n* > 1116) *	MIS-C [48] (*n* = 543)	Mild–Moderate COVID-19 [48] (*n* = 4268)	Severe COVID-19 [49,63] (*n* = 1190)
Neutrophilia	68–90%		0 =	33.2–25% =
Lymphopenia	80–95%	42.5%	1.5%	24–45% =
Diarrhea	60–100%	53.2%	3%	11.5%
Nausea/Vomiting	60–100%	57.3%	3%	17.4%
Persistent Fever	100%	97.6%	45.8%	64.3–82.7% =
Septic Shock	32–76%	21.4% (84 ^–90% #)	0.3%	13.8%
Rash	45–76%	19.5%	0.2%	0
D-dimer ↑	67–100%		3% =	12% =
Ferritin ↑	55–76%			
Fibrinogen ↑	80–100%		0 =	60% =
Troponin ↑	50–90%			2%
Procalcitonin ↑	80–95%			
C-Reactive Protein ↑	90–100%		12% =	23.5–58.7% =
Thrombocytopenia	31–80%			12% =
Met KD Criteria	22–64%		0	0
Syncope		0.2%	0	
Ischemic Stroke		0.2%	0.05–1.5% ~	16% +

**Table 2 ijms-24-03001-t002:** General overview based on sources in Introduction relating measures of innate immune system activation, possibility of co-infections with viruses and bacteria and presence of autoantibodies in various SARS-CoV-2-associated syndromes: NETs = neutrophil extracellular traps; MIS-C (PIMS) = Multisystem Inflammatory Syndrome in Children, sometimes called Pediatric Inflammatory Multisystem Syndrome Temporally Associated with SARS-CoV-2. Virus infection = SARS-CoV-2 in the COVID-19 cases or in vaccination, mild or severe COVID-19, MIS-C and COVID-19 coagulopathies, but refers to other viruses (see text) in the cases of autoimmune myocarditis and APS. Bacterial Infection = any bacterial infection epidemiologically associated with the disease listed in the left-most column. Information on autoimmune myocarditis from [81,82].

	Neutrophilia, NETs, CIC, Lymphopenia	Cytokine Over-Pro- duction	Virus Infection	Bacterial Infection	Virus– Bacterium Co-Infection	Auto- Antibodies
SARS-CoV-2 Vaccination	Extremely Rare	Extremely Rare	Rare	Rare	Extremely Rare	Rare
Mild COVID-19	Rare	Rare	Always	Rare	Rare	Rare
Severe COVID-19	Frequent	Frequent	Always	Frequent	Frequent	Frequent
MIS-C	Frequent	Frequent	Always	Frequent	Frequent	Always
Kawasaki Disease	Frequent	Frequent	Possible	Possible	Unknown	Always
Autoimmune Myocarditis	Frequent	Frequent	Frequent	Frequent	Often	Always
COVID-19 Coagulopathies	Frequent	Frequent	Always	Frequent	Frequent	Always
Anti-Phospholipid Syndrome (APS)	Frequent	Frequent	Frequent	Frequent	Probable	Always

**Table 3 ijms-24-03001-t003:** Experimentally verified evidence for the binding of antibodies against various viruses and bacteria (left-hand column) with human protein antigens associated with autoimmune coagulopathies, myopathies and vasculopathies (rows). CL = cardiolipin; β2GPI = beta 2 glycoprotein 1; PGP1b = platelet glycoprotein 1b; PT = prothrombin; F VIII = factor VIII; F IX = factor IX; vWF = von Willebrand factor; PF4 = platelet factor 4; PDE = phosphodiesterase; PL = phospholipid; Coll = collagen; SARS SP = SARS-CoV-2 spike protein; Adeno = adenovirus; Infl A = influenza A; Clost = Clostridium species; Staph = Staphylococcus aureus; Klebs = Klebsiella pneumoniae; E. coli = Escherichia coli; GAS = group A Streptococci. Blank entries indicate that there are no known reactions between human antibodies or sera to a given microbial antigen that cross-reacts with the corresponding human protein.

	CL	β2GPI	PT	F VIII	F IX	vWF	PF4	PDE	PL	Coll	Actin	Myosin
Viruses												
SARS SP						230	235, NOT 230	230	234	230, 233, 234,	233, 234	233, 234
SARS-CoV-2			230			230	230, 235	230	234	230, 233, 234,	233, 234	233, 234
Adeno		230				230	230	230		230		
Infl A												
Bacteria												
GAS	230	230	230	230	230	230	230			59, 60, 230		59, 60
E. coli	230	230					230	230				
Staph	230	230										
Klebs	230	230										
Clost.							230

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
