# Peer review of "From Co-Infections to Autoimmune Disease via Hyperactivated Innate Immunity: COVID-19 Autoimmune Coagulopathies, Autoimmune Myocarditis and Multisystem Inflammatory Syndrome in Children"

_ijms, 2023, doi:10.3390/ijms24033001_

Round 1

Reviewer 1 Report

The author carefully reviewed the involvement of pattern associated receptors in SARS-CoV-2 and its related autoimmune diseases and built up a model of explaining co-infection to autoimmune disease via hyperactivated innate immunity in sever SARS-CoV-2 cases. 

1.     Generally speaking, the review is too long, especially the introduction section, and is hard to follow. 

2.     I agree that the association of a broader involvements of TLR, NLR and RLR with autoimmune complications among severe COVID19 cases is interesting, while it should be very careful of making a cause-and-effect conclusion. To me, there is no direct evidence showing that the increased inflammation observed in COVID-19 autoimmune diseases is a result of activation of synergistic sets of innate immune system receptors by SARS-CoV-2 and bacterial antigens. SARS-CoV-2 and bacterial pathogen coinfection could be a consequence of severe SARS-CoV-2 infection alone. The subsequent dysfunctional immune system from the severe SARS-CoV-2 infection alone increases the likelihood of opportunistic infection or is susceptible to secondary bacterial super-infection. It could be explained that severe SARS-CoV-2 infection brings about self-antigen exposure and induces autoimmune responses and is independent of bacterial coinfection.   

3.     When the author evaluates whether a receptor from innate system is required for a certain disease, what is the criterion of defining that receptor is required for that disease? References that the author cited provided different levels of evidence to each receptor. As for mice work, the most straightforward way is using gene knockout approach to prove the requisite of a certain molecule. Not certain if there is a standard way of citing those references.

4.     Please check the reference that you cited is right in your paper, at lease Ref.148 is not the right reference. I know it is a lot of work. Another reason that I think the author needs to make this review concise and clear. 

Author Response

REVIEWER 1

Open Review

English language and style

( ) English very difficult to understand/incomprehensible
( ) Extensive editing of English language and style required
( ) Moderate English changes required
(x) English language and style are fine/minor spell check required
( ) I don't feel qualified to judge about the English language and style

Is the work a significant contribution to the field?

Is the work well organized and comprehensively described?

Is the work scientifically sound and not misleading?

Are there appropriate and adequate references to related and previous work?

Is the English used correct and readable?

Comments and Suggestions for Authors

The author carefully reviewed the involvement of pattern associated receptors in SARS-CoV-2 and its related autoimmune diseases and built up a model of explaining co-infection to autoimmune disease via hyperactivated innate immunity in sever SARS-CoV-2 cases. 

  1. Generally speaking, the review is too long, especially the introduction section, and is hard to follow. 

I have rewritten the Introduction to try to make it more concise and better sign-posted so that the nature of the argument and evidence that follows will be more easily followed.

  1. I agree that the association of a broader involvements of TLR, NLR and RLR with autoimmune complications among severe COVID19 cases is interesting, while it should be very careful of making a cause-and-effect conclusion. To me, there is no direct evidence showing that the increased inflammation observed in COVID-19 autoimmune diseases is a result of activation of synergistic sets of innate immune system receptors by SARS-CoV-2 and bacterial antigens. SARS-CoV-2 and bacterial pathogen coinfection could be a consequence of severe SARS-CoV-2 infection alone. The subsequent dysfunctional immune system from the severe SARS-CoV-2 infection alone increases the likelihood of opportunistic infectionor is susceptible to secondary bacterial super-infection. It could be explained that severe SARS-CoV-2 infection brings about self-antigen exposure and induces autoimmune responses and is independent of bacterial coinfection.   

The Reviewer is correct: I have over-interpreted the data in favor of bacterial co-infection. I have now taken care throughout, and especially in the Discussion, to address the possibilities of release of self-antigens and the likelihood of opportunistic or secondary super-infections.

  1. When the author evaluates whether a receptor from innate system is required for a certain disease, what is the criterion of defining that receptor is required for that disease? References that the author cited provided different levels of evidence to each receptor. As for mice work, the most straightforward way is using gene knockout approach to prove the requisite of a certain molecule. Not certain if there is a standard way of citing those references.

There is some misunderstanding here, so I have again gone over the paper to try to clarify the essential points. I did not intend to imply that certain receptors are “required for that disease” but rather that current evidence indicates that only certain receptors have reliably been established to be activated during the course of any given disease (in other words, the opposite of what the Reviewer seems to think I was saying). Thus, the reasoning is that if SARS-CoV-2 only activates TLR 3,7,9 and no NOD but severe SARS-CoV-2 is characterized by activation of TLR 2,3,4,7,9 and NOD2, then we must account for the activation of TLR2,4 and NOD2 through non-SARS-CoV-2 antigens. This reasoning does not say anything about what is “required” for the disease to be manifested but is rather about accounting for disease manifestations.  To clarify this point, I have added the statement I have just made here and also indicated its limitations, vis, that while autoimmune coagulopathies involve the activation of autoantibodies against self-antigens that do not mimic SARS-CoV-2, it is possible (as above in point 2) that release of autoantigens as a result of SARS-CoV-2-induced cellular destruction produces these additional TLR and NOD activations.

By the way, I have, as noted in the Methods, used human studies almost without exception in this paper, since the object is to explain the range of innate responses of the human immune system to the relevant diseases rather than to dissect the role of each innate receptor in these diseases. I have re-emphasized this point in the Methods and brought it up again in the Discussion as an issue that needs to be addressed in moving forward with future research.

  1. Please check the reference that you cited is right in your paper, at lease Ref.148 is not the right reference. I know it is a lot of work. Another reason that I think the author needs to make this review concise and clear. 

Thank you for pointing this out. This turned out to be a type (should have been ref 147; also now corrected in the relevant Table). Needless to say, handling the large number of references through multiple drafts involves the risk of introducing errors. I have gone over the References carefully to make sure that all the text and Table citations are now correct.

Submission Date

01 December 2022

Date of this review

19 Dec 2022 04:34:24

Reviewer 2 Report

The article ‘From Co-Infections to Autoimmune Disease via Hyperactivated Innate Immunity: COVID-19 Autoimmune Coagulopathies, Autoimmune Myocarditis, and Multisystem Inflammatory Syndrome in Children’ by Root-Bernstein summarizes the innate immune activation in severe COVID-19 and its autoimmune complications. The review article is structured well, and the work is relevant for the readership of ‘IJMS.’ However, at some places, sentences are not crafted carefully and are worded rather casually. It becomes difficult for the reader to interpret the outcome. In addition, the article does not discuss open questions and the strategies to tackle them. The authors need to address the following critical concerns in the current version of the manuscript before its publication.

The predicted synergisms and antagonisms presented in the figures appear speculative in its present form. A better approach would be to redesign these figures with color coding that highlights studies and/or predicted synergisms and antagonisms. Further, please provide a summary table with the list of literature surveyed supporting each prediction. 

While the author has attempted to summarize the relevant research in the field, not much emphasis has been given to discuss any new hypothesis, open questions, or strategies to tackle them. 

It would be helpful for the readership to have a final summary figure that delineates all the interactions discussed in the manuscript. 

At some places, the manuscript lacks tense verb consistency. Further, the authors have used long, unclear sentences, often diluting the interpretation and conclusion. These should be taken care of in the revised version of the manuscript.

Although the article is a review manuscript, the author has followed a template used for original research article. A restructuring/formatting is recommended. 

Author Response

REVIEWER 2

Open Review

English language and style

( ) English very difficult to understand/incomprehensible
( ) Extensive editing of English language and style required
(x) Moderate English changes required
( ) English language and style are fine/minor spell check required
( ) I don't feel qualified to judge about the English language and style

Is the work a significant contribution to the field?

Is the work well organized and comprehensively described?

Is the work scientifically sound and not misleading?

Are there appropriate and adequate references to related and previous work?

Is the English used correct and readable?

Comments and Suggestions for Authors

The article ‘From Co-Infections to Autoimmune Disease via Hyperactivated Innate Immunity: COVID-19 Autoimmune Coagulopathies, Autoimmune Myocarditis, and Multisystem Inflammatory Syndrome in Children’ by Root-Bernstein summarizes the innate immune activation in severe COVID-19 and its autoimmune complications. The review article is structured well, and the work is relevant for the readership of ‘IJMS.’ However, at some places, sentences are not crafted carefully and are worded rather casually. It becomes difficult for the reader to interpret the outcome.

As per Reviewer 1, I have revised the manuscript, especially the Introduction, to try to make the argument and flow of the paper more easily followed. I have added Section headings throughout and introduced additional sign-posting. I have also paid close attention to editing long and complicated sentences.

In addition, the article does not discuss open questions and the strategies to tackle them. The authors need to address the following critical concerns in the current version of the manuscript before its publication.

The predicted synergisms and antagonisms presented in the figures appear speculative in its present form. A better approach would be to redesign these figures with color coding that highlights studies and/or predicted synergisms and antagonisms. Further, please provide a summary table with the list of literature surveyed supporting each prediction. 

While the author has attempted to summarize the relevant research in the field, not much emphasis has been given to discuss any new hypothesis, open questions, or strategies to tackle them. 

The Discussion has now been expanded to address hypotheses that may explain the observations summarized in the paper and to address the open questions these raise as well as approaches to addressing them experimentally or clinically in the future.

It would be helpful for the readership to have a final summary figure that delineates all the interactions discussed in the manuscript. 

Added.

At some places, the manuscript lacks tense verb consistency. Further, the authors have used long, unclear sentences, often diluting the interpretation and conclusion. These should be taken care of in the revised version of the manuscript.

Rewritten with care taken to address these problems (as noted above).

Although the article is a review manuscript, the author has followed a template used for original research article. A restructuring/formatting is recommended. 

Submission Date

01 December 2022

Date of this review

23 Dec 2022 23:24:43

Reviewer 3 Report

This review is about the correlation between innate immunity (neutrophils) and autoimmunity and co-infection in COVID-19 patients. This review is (massive) with a lot of details which was collected by the author. He must have spent a long time reviewing this piece of work.  He explained the mediators of innate hyperactivation in severe COVID-19 and its autoimmune complications and relates these to SARS-CoV-2 activation of innate immunity.

I believe this work is not well-structured and is very difficult to follow, it is not a well-planned essay.  Several parts are unclear, and some figures are in fact (tables),  also most of them have repeated information or empty table cells.  It is confusing to follow, the author should reduce them or at least narrate his findings by different methods. The author cited his previous reviews article and used them as evidence of work. This type of citation is a secondary citation method or an indirect citation. The main source of information should not be lost by this method. Please cite original articles prior to citing your own review articles. Also, I encourage the author of this review article to re-submit after making general modifications.

Author Response

REVIEWER 3

Open Review

English language and style

( ) English very difficult to understand/incomprehensible
( ) Extensive editing of English language and style required
( ) Moderate English changes required
(x) English language and style are fine/minor spell check required
( ) I don't feel qualified to judge about the English language and style

Is the work a significant contribution to the field?

Is the work well organized and comprehensively described?

Is the work scientifically sound and not misleading?

Are there appropriate and adequate references to related and previous work?

Is the English used correct and readable?

Comments and Suggestions for Authors

This review is about the correlation between innate immunity (neutrophils) and autoimmunity and co-infection in COVID-19 patients. This review is (massive) with a lot of details which was collected by the author. He must have spent a long time reviewing this piece of work.  He explained the mediators of innate hyperactivation in severe COVID-19 and its autoimmune complications and relates these to SARS-CoV-2 activation of innate immunity.

I believe this work is not well-structured and is very difficult to follow, it is not a well-planned essay. 

This is a difficult point to address, as two other Reviewers felt that the paper was well-structured. However, given that two the Reviewers (including this one) have difficulty with the structure, I have done my best to better organize, streamline and sign-post the paper, throughout.

Several parts are unclear, and some figures are in fact (tables),  also most of them have repeated information or empty table cells.

There is a very good reason that I have listed “tables” as “figures”: my experience with this (and other) journals is that if I submit a “table”, it is always reset in a different format with italics, coloring, etc. deleted or altered. It become impossible to follow a consistent format and presentation such as that needed in the extended analysis offered here. However, if I submit the “tables” as “figures”, then the editors leave the formatting as submitted and everything remains consistent. Thus, this is not a whim on my part, but a very considered strategy to make a huge amount of data and many comparisons as easily comparable as possible across a very extended argument.

As for repeated information: of course there is! The “figure” layout is designed to show all the possible TLR-NOD interactions as a template and the filled-in boxes, those interactions that actually occur for any given disease. Not only does every TLR or NOD not interact with every other (designated by blank boxes) but the set of interactions observed in each disease differs so that different sets of filled and blank boxes create different patterns of interactions. These patterns are at the heart of the analysis presented. Since the Reviewer obviously missed this aspect of the data presentation, I have added the explanation just provided both to the “Methods” and the first “figure” caption so that the rationale cannot be missed.

 It is confusing to follow, the author should reduce them or at least narrate his findings by different methods.

As per the other Reviewers, I have attempted better to clarify the argument and data analysis throughout.

The author cited his previous reviews article and used them as evidence of work. This type of citation is a secondary citation method or an indirect citation. The main source of information should not be lost by this method. Please cite original articles prior to citing your own review articles. 

There are several issues to address here. One is that I doubt that the Reviewer would have made this comment had s/he been the author of the previous reviews. I would then have cited the reviews as “evidence of work” and I’m sure the Reviewer would not then have demanded that I cite all of the original literature that went into his/her review article. I am, in short, treating my reviews just as I would treat anyone else’s.

Secondly, I don’t know what this Reviewer considers “evidence of work” but, as far as I know, I am the only person to have taken the time and invested the energy in not just reviewing the literature on TLR-NOD interactions but also to have applied the resulting aggregate knowledge to understanding how the innate immune system responds to various infectious and autoimmune diseases. In other words, what the Reviewer calls my previous “reviews” are far more than just reviews; they go far beyond a mere listing of existing studies to create a useable framework for the knowledge and then apply that knowledge to practical medical concerns such as etiology and treatment. If that is not “evidence of work”, I’m not sure what is.

Third, none of the other three Reviewers had a problem with my citation of my previous reviews as the basis for structuring the analysis in this one.

Finally, however, I have added the entire list of studies used in my previous reviews as a Supplement to the References. Lacking context, I’m not sure how this information helps the reader but there it is.

Also, I encourage the author of this review article to re-submit after making general modifications.

Submission Date

01 December 2022

Date of this review

22 Dec 2022 08:07:08

Reviewer 4 Report

In this review the author has done a good job in reviewing the autoimmune process that occurs in COVID-19 It is interesting timely, and useful work.

Author Response

REVIEWER 4

Open Review

English language and style

( ) English very difficult to understand/incomprehensible
( ) Extensive editing of English language and style required
( ) Moderate English changes required
( ) English language and style are fine/minor spell check required
(x) I don't feel qualified to judge about the English language and style

Is the work a significant contribution to the field?

Is the work well organized and comprehensively described?

Is the work scientifically sound and not misleading?

Are there appropriate and adequate references to related and previous work?

Is the English used correct and readable?

Comments and Suggestions for Authors

In this review the author has done a good job in reviewing the autoimmune process that occurs in COVID-19 It is interesting timely, and useful work.

Thank you! Hopefully the revised version is even better.

Submission Date

01 December 2022

Date of this review

18 Dec 2022 21:25:01

Round 2

Reviewer 1 Report

This manuscript has been greatly improved after revision. 

In addition, the author has included DAMP in many places as a possible explanation. It is just not clear to me that if DAMP induced RPRs overlay with bacterial induced RPRs? I think some are. As from this paper, DAMP activated RPRs including TLR2, TLR4, TLR6, NLRP31. Then it is possible that at least some of the additional receptors that were only identified in severe SARS-COV-2 cases, but not in mild cases maybe activated by DAMP. When you created the analytical framework of receptor synergisms in this review, would you be able to differentiate those overlayed receptors between bacterial origin or DAMP origin? If not, I think it worth to point out the limitation of this model, although I think the author has mentioned it a bit in the manuscript.

1.     Schaefer L. Complexity of danger: the diverse nature of damage-associated molecular patterns[J]. Journal of Biological Chemistry, 2014, 289(51): 35237-35245.

Author Response

REVIEWER 1

Open Review

English language and style

( ) English very difficult to understand/incomprehensible
( ) Extensive editing of English language and style required
( ) Moderate English changes required
(x) English language and style are fine/minor spell check required
( ) I don't feel qualified to judge about the English language and style

Is the work a significant contribution to the field?

Is the work well organized and comprehensively described?

Is the work scientifically sound and not misleading?

Are there appropriate and adequate references to related and previous work?

Is the English used correct and readable?

Comments and Suggestions for Authors

This manuscript has been greatly improved after revision. 

In addition, the author has included DAMP in many places as a possible explanation. It is just not clear to me that if DAMP induced RPRs overlay with bacterial induced RPRs? I think some are. As from this paper, DAMP activated RPRs including TLR2, TLR4, TLR6, NLRP31. Then it is possible that at least some of the additional receptors that were only identified in severe SARS-COV-2 cases, but not in mild cases maybe activated by DAMP. When you created the analytical framework of receptor synergisms in this review, would you be able to differentiate those overlayed receptors between bacterial origin or DAMP origin? If not, I think it worth to point out the limitation of this model, although I think the author has mentioned it a bit in the manuscript.

 NEW PARAGRAPH AND FIGURE 12 ADDED TO END OF DISCUSSION (HIGHLIGHTED IN YELLOW) ALONG WITH THE FOLLOWING REFRENCE AND THREE ADDITIONAL REVIEWS OF DAMP ACTIVATION PATTERNS OF TLR AND NLR.

  1. Schaefer L. Complexity of danger: the diverse nature of damage-associated molecular patterns[J]. Journal of Biological Chemistry, 2014, 289(51): 35237-35245.

Submission Date

01 December 2022

Date of this review

15 Jan 2023 16:49:29

Reviewer 2 Report

After careful examination of the revised manuscript, the response of the authors to previous reviews, and the changes made in the manuscript, I gather that the revised version of the manuscript has addressed the major concerns raised in the previous version of the paper. Hence, I endorse the publication of this paper.

Author Response

REVIEWER 2

Open Review

English language and style

( ) English very difficult to understand/incomprehensible
( ) Extensive editing of English language and style required
( ) Moderate English changes required
(x) English language and style are fine/minor spell check required
( ) I don't feel qualified to judge about the English language and style

Is the work a significant contribution to the field?

Is the work well organized and comprehensively described?

Is the work scientifically sound and not misleading?

Are there appropriate and adequate references to related and previous work?

Is the English used correct and readable?

Comments and Suggestions for Authors

After careful examination of the revised manuscript, the response of the authors to previous reviews, and the changes made in the manuscript, I gather that the revised version of the manuscript has addressed the major concerns raised in the previous version of the paper. Hence, I endorse the publication of this paper.

THANK YOU FOR THE USEFUL SUGGESTIONS!

Submission Date

01 December 2022

Reviewer 3 Report

Dear Author

The work has been much improved. 

Thanks for your collaborating

Author Response

REVIEWER 3

Open Review

English language and style

( ) English very difficult to understand/incomprehensible
( ) Extensive editing of English language and style required
( ) Moderate English changes required
( ) English language and style are fine/minor spell check required
(x) I don't feel qualified to judge about the English language and style

Is the work a significant contribution to the field?

Is the work well organized and comprehensively described?

Is the work scientifically sound and not misleading?

Are there appropriate and adequate references to related and previous work?

Is the English used correct and readable?

Comments and Suggestions for Authors

Dear Author

The work has been much improved. 

Thanks for your collaborating

 AND THANK YOU!

Submission Date

01 December 2022

Date of this review

04 Jan 2023 09:28:32